# Kv1.1 channels regulate early postnatal neurogenesis in mouse hippocampus via the TrkB signaling pathway

Shu-Min Chou[1†‡], Ke-Xin Li[2†], Ming-Yueh Huang[3†], Chao Chen[2],
Yuan-Hung Lin King[2,4], Grant Guangnan Li[5], Wei Zhou[6], Chin Fen Teo[2],
Yuh Nung Jan[2], Lily Yeh Jan[2]*, Shi-Bing Yang[1,7]*

[1]Institute of Biomedical Sciences, Academia Sinica, Taipei, Taiwan; [2]Howard Hughes Medical Institute, Departments of Physiology, Biochemistry and Biophysics, University of California, San Francisco, San Francisco, United States; [3]Institute of Statistics, Academia Sinica, Taipei, Taiwan; [4]Neuroscience Graduate Program, University of California, San Francisco, San Francisco, United States; [5]Nkarta Therapeutics Inc, South San Francisco, United States; [6]Department of Anesthesia and Perioperative Care, University of California, San Francisco, San Francisco, United States; [7]Neuroscience Program of Academia Sinica, Academia Sinica, Taipei, Taiwan

*For correspondence:
Lily.Jan@ucsf.edu (LYJ);
sbyang@ibms.sinica.edu.tw (S-BY)

[†]These authors contributed equally to this work

Present address: [‡]Program in Neuroscience and Behavioral Disorders, Duke-National, University of Singapore Medical School, Singapore, Singapore

**Abstract** In the postnatal brain, neurogenesis occurs only within a few regions, such as the hippocampal sub-granular zone (SGZ). Postnatal neurogenesis is tightly regulated by factors that balance stem cell renewal with differentiation, and it gives rise to neurons that participate in learning and memory formation. The Kv1.1 channel, a voltage-gated potassium channel, was previously shown to suppress postnatal neurogenesis in the SGZ in a cell-autonomous manner. In this study, we have clarified the physiological and molecular mechanisms underlying Kv1.1-dependent postnatal neurogenesis. First, we discovered that the membrane potential of neural progenitor cells is highly dynamic during development. We further established a multinomial logistic regression model for cell-type classification based on the biophysical characteristics and corresponding cell markers. We found that the loss of Kv1.1 channel activity causes significant depolarization of type 2b neural progenitor cells. This depolarization is associated with increased tropomyosin receptor kinase B (TrkB) signaling and proliferation of neural progenitor cells; suppressing TrkB signaling reduces the extent of postnatal neurogenesis. Thus, our study defines the role of the Kv1.1 potassium channel in regulating the proliferation of postnatal neural progenitor cells in mouse hippocampus.

## Introduction

In mammals, the majority of neurons in the central nervous system are generated during embryonic development, and postnatal neurogenesis is limited to only a few brain regions, such as the subgranular zone (SGZ) of the dentate gyrus and the sub-ventricular zone (SVZ) of the lateral ventricles (*Kriegstein and Alvarez-Buylla, 2009*). Neural progenitor cells in the SGZ are categorized into different developmental stages based on distinct cellular functions, and cells of each stage can be identified by the expression of specific cell-fate markers. For example, type 1 radial glia-like neural stem cells have the potential for self-renewal and mostly stay quiescent; these cells are positive for Brain Lipid Binding Protein (BLBP), Glial Fibrillary Acidic Protein (GFAP), and nestin. Another widely used neural stem cell marker is Sox2, a transcription factor that is essential for maintaining stem cell pluripotency. Type 1 cells undergo asymmetric cell division to generate highly proliferative type 2a neural

progenitor cells, which are double-positive for Sox2 and Tbr2, transcription factors that control cell proliferation and differentiation. Type 2a cells then generate proliferative but lineage-restricted type 2b neural progenitor cells, which express Tbr2 and doublecortin (DCX) (*Gonçalves et al., 2016*; *Ming and Song, 2011*; *Spampanato et al., 2012*). Type 2b cells exhibit limited mitotic potential (*Gonçalves et al., 2016*), and after two to five rounds of mitosis, these progenitor cells differentiate into immature neurons (*Gao et al., 2014*). In the adult SGZ, the newly generated granule cells contribute to memory formation, likely by facilitating the ability to distinguish similar patterns, a learning process referred to as pattern separation (*Sahay et al., 2011*). Importantly, postnatal neurogenesis is tightly regulated by the balance between self-renewal and differentiation (*Ming and Song, 2005*). While the regulation of these processes by intrinsic transcription factors has been studied extensively (*Aimone et al., 2014*), the impacts of membrane potential and ion channels on postnatal neurogenesis are much less clear (*Fukuda et al., 2003*; *Swayne and Wicki-Stordeur, 2012*).

Ion channels are expressed in almost all cell types, including highly proliferative stem cells and cancers; however, many ion channel-expressing cell types are considered to be electrically non-excitable (*Bates, 2015*; *Fukuda et al., 2003*; *Shin et al., 2015*). The type 1 stem cells in the SGZ display an immense passive conductance comprised of connexin-based gap junctions (*Kunze et al., 2009*; *Rozental et al., 1998*) as well as Kir4.1- and Kir5.1-mediated glial-type inwardly rectifying potassium channels (*Yasuda et al., 2008*). This passive conductance renders neural stem cells unable to fire action potentials (*Fukuda et al., 2003*; *Swayne and Wicki-Stordeur, 2012*). In addition to the passive conductance, neural stem cells and progenitor cells also exhibit various features of active conductance, such as voltage-gated potassium channels (*Bates, 2015*; *Shin et al., 2015*), and to a lesser extent, calcium channels (*D'Ascenzo et al., 2006*; *Xu et al., 2018*). One such voltage-gated potassium channel expressed in the embryonic brain and postnatal neural stem cells is Kv1.1 (*Hallows and Tempel, 1998*; *Shin et al., 2015*). This channel is found mainly in mature neurons (*Wang et al., 1994*), and it is predominantly localized in axons due to microtubule End-Binding 1 (EB1)-directed targeting (*Gu et al., 2006*; *Hallows and Tempel, 1998*). Functionally, Kv1.1 is best-known for its role in tuning action potential firing patterns by regulating the duration of the repolarization phase (*Foust et al., 2011*; *Jan and Jan, 2012*; *Storm, 1988*). Mutations of the Kv1.1 channel cause various neurological disorders, such as epilepsy and episodic ataxia, in humans and mouse models (*Beraud et al., 2006*; *Heeroma et al., 2009*; *Petersson et al., 2003*; *Robbins and Tempel, 2012*). Moreover, since Kv1.1 can be activated at a relatively low voltage (near the resting potential), it has been suggested that this potassium channel could modulate membrane electrical properties such as the membrane potential in various cell types, including non-excitable neural progenitor cells (*Foust et al., 2011*; *Storm, 1988*; *Sundelacruz et al., 2013*).

Kv1.1 channel, the product of the gene *Kcna1*, is thought to regulate postnatal neurogenesis since megencephaly mice (*mceph*) lacking functional Kv1.1 channels have enhanced adult neurogenesis that leads to enlargement of the hippocampi (*Persson et al., 2007*). However, the mechanism of this action is largely unknown. Many ion channels are widely expressed in the central nervous system, and their mutations may either directly influence both neural progenitor cells and their progeny neurons (cell-autonomous) or indirectly affect neural progenitor cells by modulating the excitability of mature neurons (non-cell-autonomous) (*Espinosa et al., 2009*; *Piggott et al., 2019*). Whereas seizures resulting from the loss of Kv1.1 function may impact postnatal neurogenesis (*Almgren et al., 2008*; *Holth et al., 2013*), our previous study using mosaic analysis with double markers (MADM) (*Muzumdar et al., 2007*; *Zong et al., 2005*) on heterozygous *mceph* mice without seizures revealed that Kv1.1 regulates postnatal neurogenesis via a cell-autonomous mechanism (*Yang et al., 2012a*). To avoid the potentially confounding effects of seizures, which start by 3–4 weeks of age in Kv1.1-null (*Kcna1$^{-/-}$*) mice, we focused on early postnatal neurogenesis in the current study. We first showed that Kv1.1 affects early postnatal neurogenesis without having a detectable impact on embryonic neurogenesis. Then, we searched for the molecular and physiological mechanisms that mediate the Kv1.1-dependent postnatal neurogenesis in mouse hippocampus. Our results indicated that Kv1.1 cell-autonomously modulates the membrane excitability of type 2b neural progenitor cells and TrkB signaling, which accounts for increased proliferating neural progenitor cells in Kv1.1KO mice.

## Results

### Kv1.1 regulates postnatal neuron generation in a cell-autonomous manner

The Kv1.1 channel is widely expressed throughout the entire central nervous system (*Hallows and Tempel, 1998*), and the loss of Kv1.1 function in mice homozygous for either the *megencephaly* mutation or the Kv1.1-null mutation (Kv1.1KO) causes seizures, starting 3 weeks after birth (*Donahue et al., 1996*; *Petersson et al., 2003*). Our previous study in 3-month-old *Kcna1*^mceph/+ mice carrying the MADM-6 b cassettes (*Rosa26*^TG/GT) showed that Kv1.1 acts cell-autonomously to regulate the number of neuronal progeny (*Yang et al., 2012a*). We further carried out the MADM study in *Kcna1*⁻ mice, which did not show signs of seizures before reaching adulthood (*Figure 1*). We first generated Nestin-cre;*Kcna1*; MADM-6 quadruple transgenic mice that carried Nestin-cre, *Kcna1*⁻, *Rosa26*^TG, and *Rosa26*^GT. Nestin-Cre mediated somatic recombination in a subset of neural progenitor cells that carried the MADM-6 cassettes, and the daughter cells bearing the *Kcna1*^-/- alleles were labeled with green fluorescent protein (GFP) (green), while the wild-type sibling daughter cells were labeled with tdTomato (red) (*Figure 1A,B,C,D*). Comparable numbers of green Kv1.1KO neurons and red wild-type neurons were observed in 1-month-old Nestin-cre;*Kcna1*^+/-; MADM-6 mice; however, a significantly larger number of green Kv1.1KO neurons was found in 2- to 3-month-old Nestin-cre;*Kcna1*^+/-;MADM-6 mice compared to wild-type controls (*Figure 1E*). As expected for granule cells born postnatally, the newly generated green Kv1.1KO neurons were located farther from the SGZ in the granule cell layer (*Figure 1F*). Unlike adult-born neurons, which gradually decrease in number over the course of a month owing to apoptosis, developmentally-born neurons generated during late embryogenesis and early postnatal stages typically survive for more than 2 months before the onset of cell death; generation and maturation of these neurons take about 1 month, with those generated early in life located farther from the SGZ (*Cahill et al., 2017*; *Dayer et al., 2003*; *Kerloch et al., 2019*; *Toni and Schinder, 2015*). Thus, the MADM experiments revealed an overproduction of developmentally born neurons from Kv1.1KO neural progenitor cells in early postnatal stages.

To test whether Kv1.1 function is essential for neurogenesis during embryogenesis, we examined embryos with or without Kv1.1 and found no significant difference in the numbers of hippocampal progenitors or neurons at E16.5 (*Figure 1—figure supplement 1*). We also analyzed interneurons expressing parvalbumin (PV) or somatostatin (SST) in the dentate gyrus and found no significant difference between mice with or without Kv1.1 at P10 (*Figure 1—figure supplement 2*). Taken together, our experiments indicate that Kv1.1 functions in postnatal neural progenitors of the SGZ to regulate the production of neuronal progeny during the first 3 months of life, raising the question regarding the mechanisms underlying this process.

### Functional expression of Kv1.1 in SGZ neural progenitor cells

Kv1.1 is highly expressed throughout the entire central nervous system (*Wang et al., 1994*). Given that neural progenitor cells only account for a small fraction of total cells in the brain, we decided to utilize Fezf2-GFP reporter mice to specifically examine neural stem cells and progenitor cells in the SGZ (*Gong et al., 2003*). The majority of the Fezf2-GFP-positive cells in the SGZ co-expressed BLBP and GFAP, two well-established neural stem cell markers (*Berberoglu et al., 2014*). We also found that Kv1.1 mRNA was expressed in the granule cell layer of the dentate gyrus (red dots, *Figure 2A*); a portion of the Kv1.1 mRNA signal was in close proximity to the GFP mRNA (green dots), which indicates Fezf2-GFP neural progenitor cells in the SGZ (*Figure 2A,B*). To further corroborate this finding, we performed immunostaining to determine the distribution pattern of the Kv1.1 protein in the mouse dentate gyrus. We found that Kv1.1 protein was highly expressed in mature granule cells and interneurons as reported previously (*Grosse et al., 2000*; *Wang et al., 1994*). A closer look further revealed the presence of Kv1.1 in DCX+ cells but not in Sox2+ cells, thereby confirming the expression of Kv1.1 in late-stage neural progenitor cells (*Figure 2C*).

Next, we wanted to determine whether Kv1.1 protein could form functional potassium channels in the Fezf2-GFP-positive cells. As Kv1.1-mediated potassium currents can be inactivated by holding at relatively high voltages (> −50 mV) (*Storm, 1988*) and are sensitive to dendrotoxin-K (DTX-K) (*Grissmer et al., 1994*), a potent blocker of Kv1 family channels, we performed whole-cell patch-

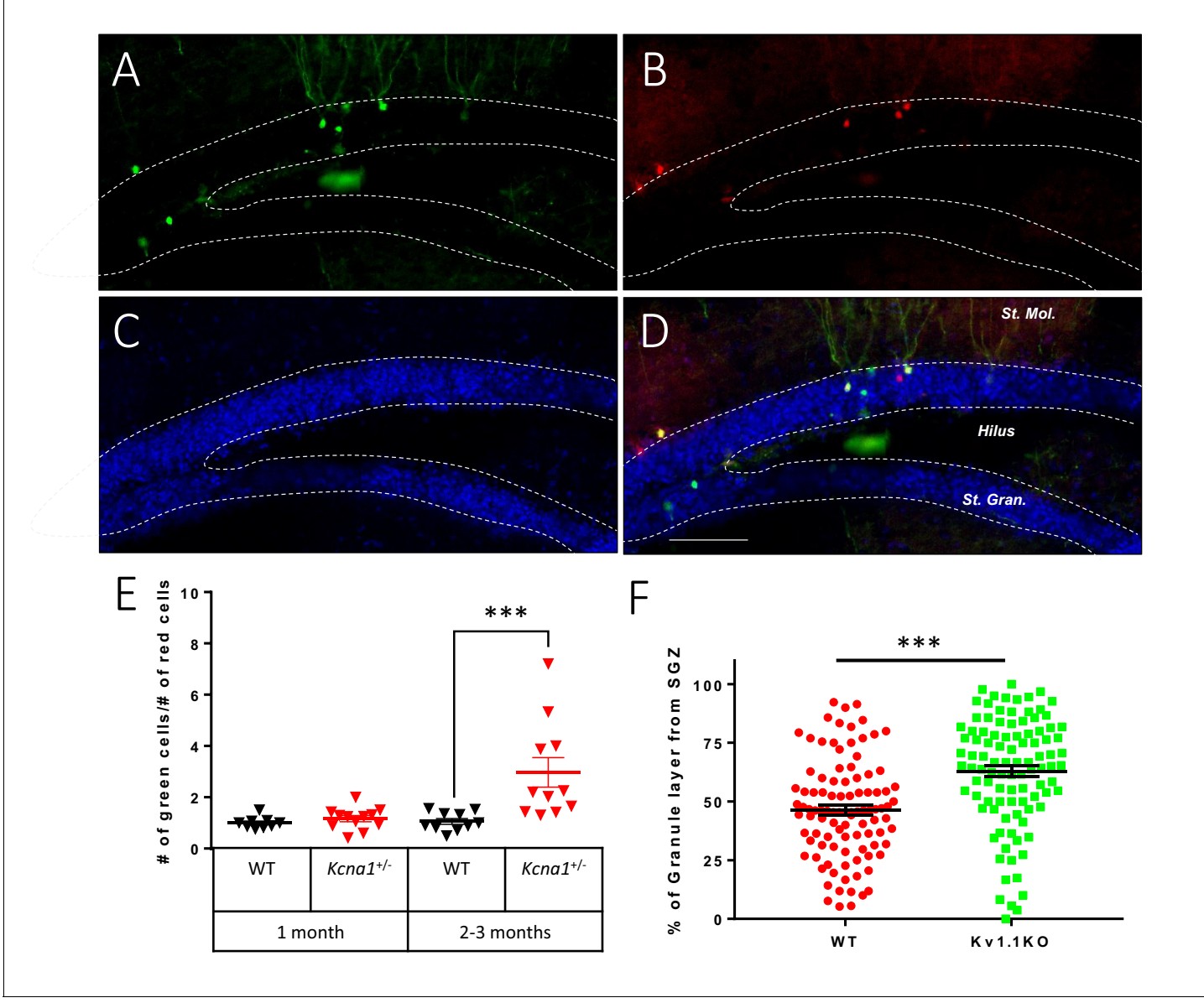

**Figure 1.** Kv1.1 channels control dentate gyrus neuron number in a cell-autonomous manner. Mosaic analysis with double markers (MADM) of the dentate gyrus in wild-type or *Kcna1*$^{+/-}$ mice. Kv1.1KO and wild-type neurons can be respectively identified as green (GFP) (**A**) and red (TdTomato) (**B**) neurons. (**C**) 4′,6-diamidino-2-phenylindole (DAPI) counterstain. (**D**) Overlay of signals in (**A**) and (**B**), scale bar = 100 μm. (**E**) Statistical analysis of the red wild-type and green Kv1.1KO neurons in the dentate gyrus. The numbers of red wild-type and green Kv1.1KO neurons were comparable in the dentate gyrus of 1-month-old mice; however, compared to the numbers of red wild-type neurons, the numbers of green Kv1.1KO neurons were much higher in the dentate gyrus of 2- to 3-month-old mice (two-way ANOVA followed by Sidak's multiple comparisons test; p=0.0008, for 2- to 3-month-old mice). n = 9–12 for each group. (**F**) Compared to the red wild-type neurons, green Kv1.1KO neurons were more often positioned farther away from the sub-granular zone (SGZ) in 2- to 3-month-old mice, indicating that the green Kv1.1KO neurons were born in the postnatal period (p=0.0002, Mann-Whitney *U*-test). n = 100 for each group.

The online version of this article includes the following figure supplement(s) for figure 1:

**Figure supplement 1.** Kv1.1 channels do not affect embryonic hippocampal neurogenesis.

**Figure supplement 2.** No detectable difference in interneurons of Kv1.1KO mice at P10.

clamp recording in wild-type (*Figure 2D,E,F*) and Kv1.1KO (*Figure 2F,G,I*) Fezf2-GFP-positive cells. We could detect DTX-K-sensitive and low-voltage inactivated potassium currents in wild-type but not Kv1.1KO (*Figure 2J*) Fezf2-GFP+ cells. These findings reveal that Kv1.1 forms functional channels in Fezf2-GFP neural progenitor cells.

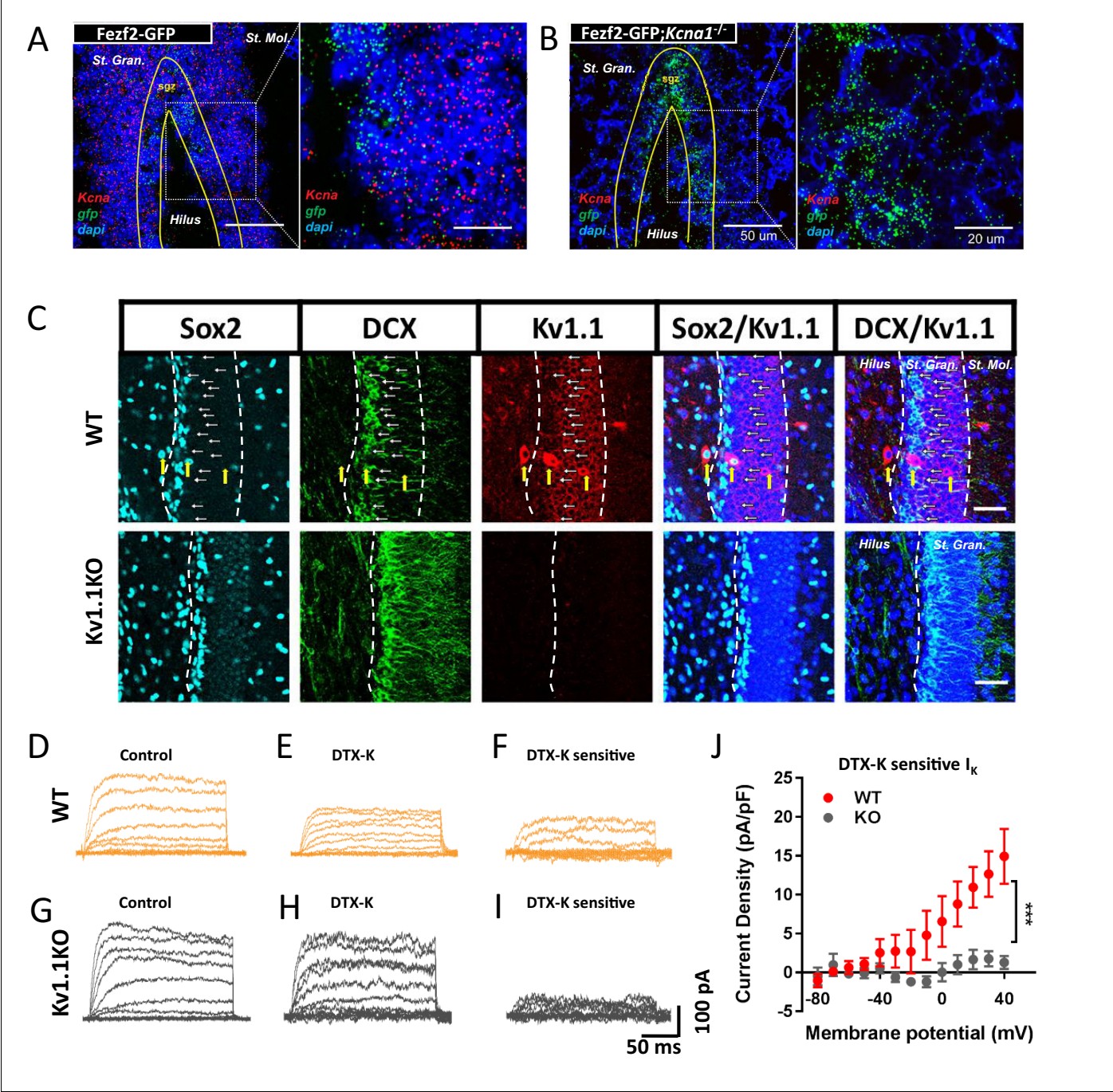

**Figure 2.** Functional expression of Kv1.1 channel in SGZ neural progenitor cells expressing Fezf2-GFP at postnatal 2 weeks. (**A**) In situ hybridization results showed that *Kcna1* mRNA is expressed in Fezf2-GFP-positive neural progenitor cells. Inset is displayed at a higher magnification. (**B**) *Kcna1* mRNA was not detected in the Kv1.1KO mouse, which served as a negative control. (**C**) Kv1.1 protein was expressed in the doublecortin (DCX)-expressing late-stage neural progenitor cells (arrows) but not Sox2-positive early-stage neural progenitor cells. Kv1.1 protein was highly expressed in the inhibitory interneurons (yellow arrows). (**D–J**) Pharmacological isolation of Kv1 currents in Fezf2-GFP-positive cells. Kv1.1 currents were elicited by trains of voltage steps from -80 mV to +40 mV in 10 mV increments from the holding potential of -80 mV in the absence (**D and G**) or presence of the Kv1-specific blocker dendrotoxin-k (DTX-K; 100 nM) (**E and H**). The DTX-K-sensitive currents were considered Kv1-mediated potassium currents (**F, I, and J**), which were much reduced in the Kv1.1KO mice. n = 4 cells from each phenotype. Scale bar = 20 μm in (**C**). Data are presented as mean ± SEM.

## Loss of Kv1.1 depolarizes a subset of the neural progenitor cells in SGZ

Next, we looked into the impact of removing Kv1.1 from Fezf2-GFP cells at different developmental stages (*Figure 3A*). This experiment was performed within the first postnatal month to avoid complications from seizures that begin later in life. We sought to evaluate quiescent Sox2+ type 1 radial glia-like stem cells, highly proliferative Sox2+Tbr2+ type 2a neural progenitor cells, and type 2b neural progenitor cells that only express Tbr2. Based on the expression of cell-type markers, we first calculated the proportions of these three cell types among Fezf2-GFP cells. We found a high proportion of Tbr2+-only type 2b neural progenitor cells in Kv1.1KO SGZ compared to wild type (*Figure 3B,C*, arrowheads). These results suggested that the presence of Kv1.1 limits the number of late-stage neural progenitor cells.

Voltage-gated potassium channels are known to be involved in regulating action potential waveforms and firing frequencies, but this type of channel plays only minor roles in tuning passive properties such as the resting membrane potential (*Corbin-Leftwich et al., 2018*). For example, neurons lacking functional Kv1.1 display normal resting potentials but elevated action potential firing rates (*Robbins and Tempel, 2012*; *Smart et al., 1998*). To test whether the loss of Kv1.1 affects the membrane potential of neural progenitor cells, we recorded from Fezf2-GFP cells and POMC-GFP immature neurons, as the Pomc-GFP transgenic line faithfully labels immature neurons in the dentate gyrus (*Overstreet et al., 2004*). In addition, a previous study has shown that the radial glia-like cells are coupled by gap junctions that are essential for adult neurogenesis in the SGZ (*Kunze et al.,*

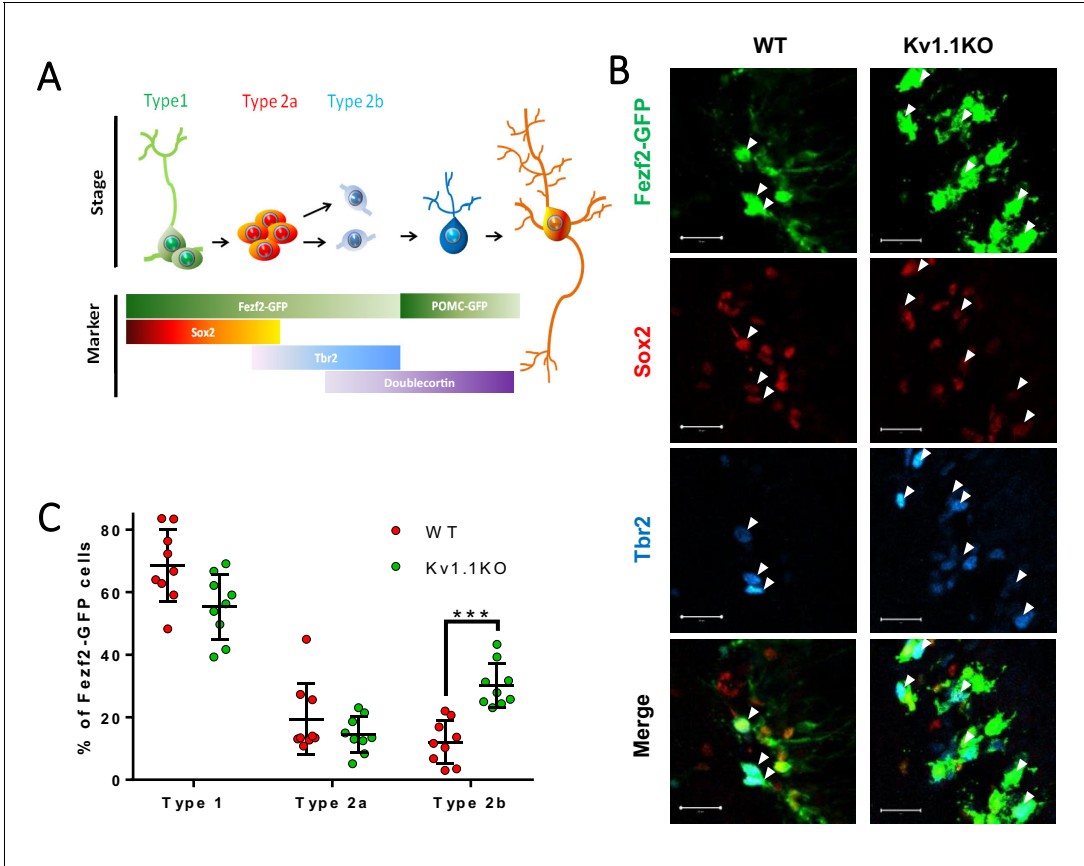

**Figure 3.** Loss of Kv1.1 channels depolarizes type 2b neural progenitor cells in SGZ at postnatal 2 weeks. (**A**) Based on the expression of the cell-fate markers, the neural progenitor cells in the sub-granular zone (SGZ) can be categorized into several developmental stages (see *Figure 4* for detailed images). (**B**) Among all the Fezf2-GFP-positive cells in the SGZ, radial glia-like type 1 neural progenitor cells expressed predominantly Sox2, type 2a neural progenitor cells expressed both Sox2 and Tbr2 (white arrows), and type 2b progenitor cells expressed Tbr2. Post-mitotic immature neurons expressed doublecortin (DCX) and could also be identified as POMC-GFP-positive cells. (**C**) Among these cell types, type 2b cells were significantly increased in adult Kv1.1KO dentate gyrus compared to wild type (WT), according to the numbers of Tbr2-positive Fezf2-GFP-positive cells (n = 9 for each group; two-way ANOVA followed by Sidak's multiple comparisons test; p=0.0009, for type 2b cells). Scale bar = 20 µm.

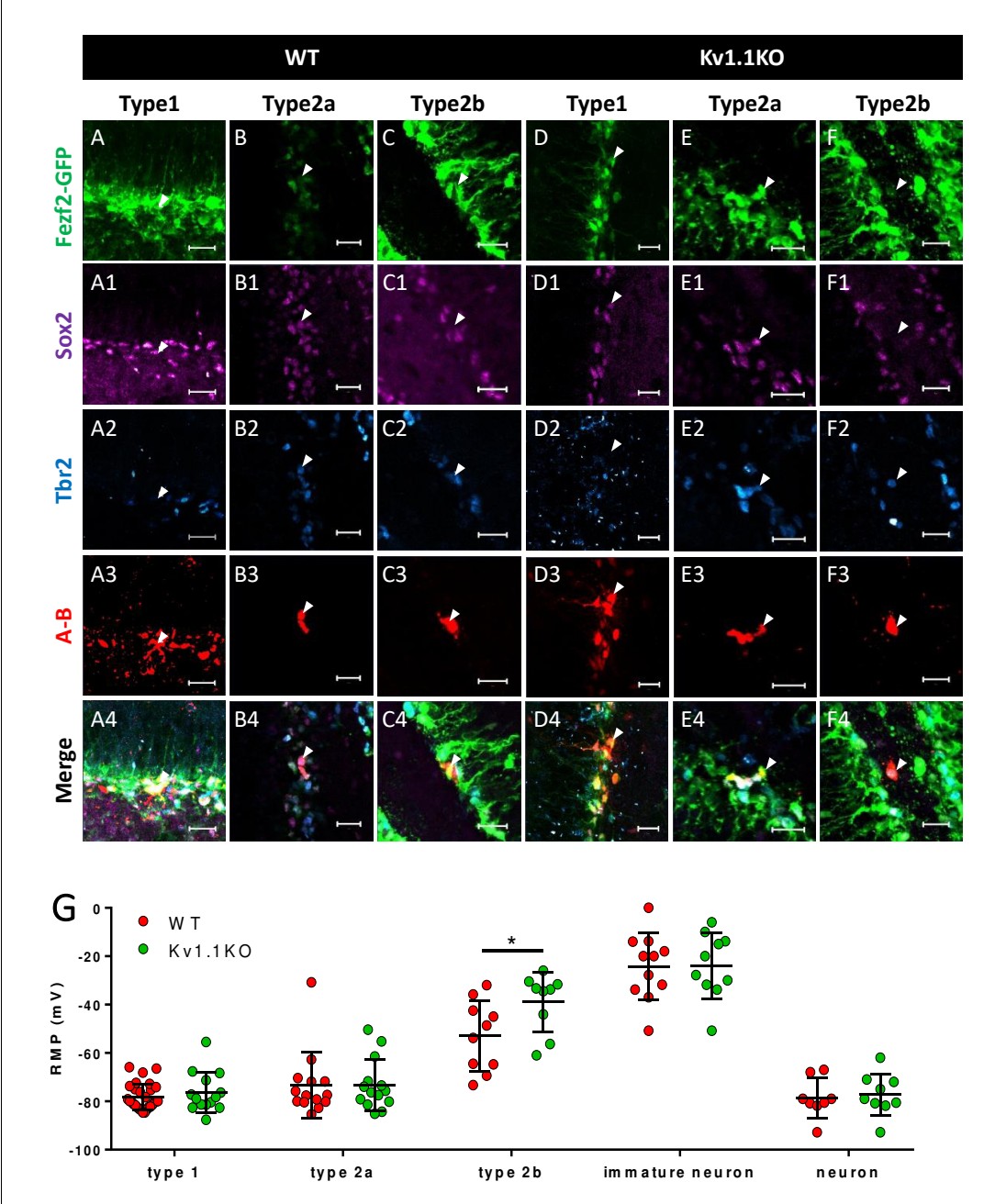

**Figure 4.** Classification of Fezf2-GFP progenitor cells in SGZ. Based on the expression of the cell-fate markers, neural progenitor cells in the sub-granular zone (SGZ) can be categorized into several developmental stages. Among all the Fezf2-GFP-positive cells (A–F), Sox2-positive type 1 cells (A1–F1) are radial glia-like cells (A–A4, C–C4) and can give rise to Sox2+/Tbr2+ type 2a transit-amplifying progenitor cells (B–B4, D–D4). Type 2a cells further differentiate into type 2b neuroblasts that are positive for Tbr2 only (C–C4, F–F4). Type 1 (A3 and D3) and type 2a (B3 and E3) cells formed extensive syncytial connections with other cells from the same developmental stage, as indicated by the gap-junction-permeable avidin-neurobiotin (A–B) staining. Scale bar = 20 μm. (G) Type 1 and type 2a Fezf2-GFP-positive neural progenitor cells were hyperpolarized; by contrast, type 2b cells lacking Kv1.1 channels were significantly more depolarized than the wild-type (WT) cells (n = 14, 14, 9, 10, 9 (WT) and 28, 14, 10, 11, 8 (Kv1.1KO) for type 1 cells (Fezf2-GFP+/Sox2+), type 2a cells (Fezf2-GFP+/Sox2+/Tbr2+), type 2b cells (Fezf2-GFP+/Tbr2+), immature neurons (POMC-GFP+), and label-free mature neurons; (two-way ANOVA followed by Sidak's multiple comparisons test; p=0.02, for type 2b cells)).

*2009*). In order to label the recorded neural progenitor cells in brain slices, we included a gap-junction-permeable tracer neurobiotin in the pipette solution (*Figure 4*, A3–F3 and *Figure 5A*). After whole-cell patch-clamp recording to characterize the membrane properties, the brain slices were fixed with 4% paraformaldehyde (PFA). The cell-type identity of the recorded cell was characterized by examining the expression of cell-fate markers (*Figure 3A*). The membrane potentials of these cells were hyperpolarized and electrically silent; notably, the current injection could cause depolarization of POMC-GFP immature neurons but not firing of action potentials, while such depolarization of mature granule neurons could lead to action potential firing (data not shown). Remarkably, the Tbr2+/Fezf2+ type 2b neural progenitor cells lacking Kv1.1 displayed more depolarized resting membrane potentials than their wild-type counterparts (*Figure 4G*). Moreover, we found that type 2a progenitor cells were coupled via gap junctions (*Figure 4B,E*), a property that has not been reported previously.

Although we have obtained convincing results showing that the type 2b neural progenitor cells lacking Kv1.1 channels were more depolarized, the invasive nature of the patch-clamp technique caused more than 70% of the recorded cells to be either damaged or lost during the subsequent immunostaining procedure, and therefore, their developmental stages were not determined. We tried to classify those Fezf2-GFP-positive cells based on their resting membrane potential, input resistance, and membrane capacitance, the three basic biophysical characteristics obtained from patch-clamp recording (*Figure 5*). First, we used the biophysical characteristics of cells with unequivocal cell-type identification determined by post-hoc immunostaining as a training dataset (*Figure 4G*) to construct separate multinomial logistic regression models for wild-type and Kv1.1KO cells (*Fahrmeir and Tutz, 2001*). The probability of each cell belonging to a given cell type was calculated by fitting biophysical characteristics to the models: resting membrane potentials (*Figure 5D*), input resistance (*Figure 5E*), and membrane capacitance (*Figure 5F*). We found that more than 80% of the Fezf2-GFP-positive cells with known cell types could be accurately predicted using this regression model. Interestingly, judging from the gap-junction-permeable neurobiotin labeling, we noticed that cells from the same subtypes clustered closely together (*Figure 5B, C*). Of note, type 1 neural stem cells tended to have a more hyperpolarized resting membrane potential, lower input resistance, and larger membrane capacitance, while type 2b neural progenitor cells tended to have more depolarized resting membrane potentials, higher input resistance, and smaller membrane capacitance. The cell type was then assigned based on the highest probability. Consistent with the results shown in *Figure 4G*, we found that the assigned type 2b neural progenitor cells lacking Kv1.1 channel were also significantly more depolarized than their wild-type counterparts. Nevertheless, both the input resistance (*Figure 5E*) and membrane capacitance (*Figure 5F*) were comparable between wild-type and Kv1.1KO cells. To validate the progenitor classification and reveal the cell membrane properties, we plotted *I-V* curves for the three subgroups, as classified by the multinomial logistic regression model. Similar to the results of a previous study using progenitor cell recording (*Steiner et al., 2006*), type 1 neural stem cells characteristically displayed passive, non-inactivating currents with a linear current–voltage relationship (*Figure 5J,M*), similar to the properties of astrocytes. In contrast, type 2b neural progenitor cells expressed small outwardly rectifying currents (*Figure 5L,M*). Of note, in 9 out of 25 type 2b neural progenitor cells, we observed transient inward currents induced by depolarizations more positive than −30 mV, followed by outwardly rectifying currents; these observations are indicative of sodium currents, hence consistent with neuronal differentiation (*Figure 5P*). The delayed outward rectifying currents in type 2b neural progenitor cells were more apparent when characterized by means of leak subtraction (*Figure 5P,O*). Moreover, the electrophysiological features of type 2a neural progenitor cells were suggestive of a transition phase between type 1 neural stem cells and type 2b neural progenitor cells, since type 2a neural progenitor cells displayed outwardly rectifying currents but no detectable sodium currents (*Figure 5K,M*).

## Kv1.1 regulates neural progenitor cell number via the TrkB signaling pathway

A previous work has shown that suppressing TrkB signaling by removing brain-derived neurotrophic factor (BDNF) from neural progenitor cells drastically reduces the thickness of the dentate gyrus granule cell layer (*Li et al., 2008*), while enhancing TrkB signaling by overexpressing BDNF in neural progenitor cells augments neurogenesis (*Quesseveur et al., 2013*). Notably, increased BDNF

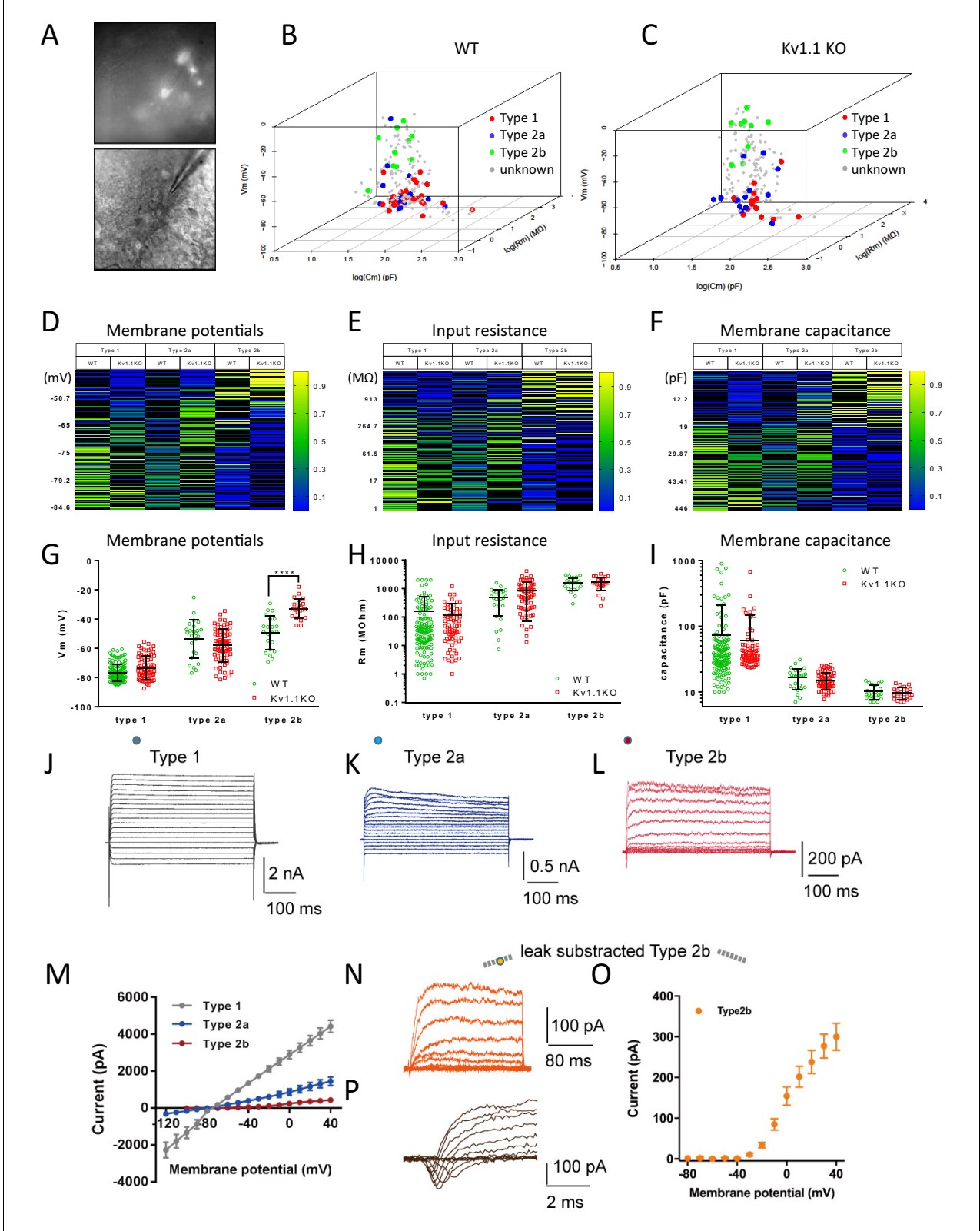

**Figure 5.** Multinomial logistic regression for cell-type prediction. (**A**) Acute brain slice recording from the DG of an Fezf2-GFP mouse. (**B and C**) A multinomial logistic regression model was constructed using the biophysical characteristics of cells that were identified previously (same cohort of cells from *Figure 3D*) as the training dataset. Two individual regression models were generated for wild-type (**B**) and Kv1.1KO (**C**) cells. The cell types of unknown cells were later classified according to the resting membrane potential, input resistance, and membrane capacitance. Heatmaps show the

*Figure 5 continued on next page*

Figure 5 continued

estimated probabilities for each cell type against membrane potential (D), input resistance (E), and membrane capacitance (D). The cell types were determined based on maximum likelihood. (G) Resting membrane potentials of wild-type and Kv1.1KO progenitor cells. The predicted type 2b cells lacking Kv1.1 channels were significantly more depolarized (two-way ANOVA, followed by Sidak's multiple comparisons test; $p<0.0001$, for type 2b cells). Comparable input resistances (H) and membrane capacitances (I) were observed between wild-type and Kv1.1KO cells. n = 144, 25, and 22 for type 1, 2a, and 2b cells, respectively, in the wild-type group; n = 71, 70, and 22 for type 1, 2a, and 2b cells, respectively, in the Kv1.1KO group. (J–L) Voltage responses from type 1 (J), type 2a (K), type 2b (L) to series voltage from -120 mV to +40 mV or -80 mV to +40 mV, with a holding potential of -80 mV. (M) Current–voltage curves in (J–L) (the circles indicate the positions of measurements; type 1 (7 cells), type 2a (7 cells), type 2b (10 cells)). (N, O) Voltage response from -80 mV to +40 mV, with a holding potential of -80 mV and leak subtraction, shown for type 2b (N); current–voltage curve of type 2b (n = 17 cells) with leak subtraction (O). (P) A representative transient inward current (sodium current) was observed in 9 out of 25 type 2b cells. Data are presented as mean ± SD in (G), (H), and (I); mean ± SEM in (M, O).

The online version of this article includes the following source data for figure 5:

**Source data 1.** multiple logistic regression datasets for biophysical analysis of the neural progenitor cells in postnatal mouse hippocampus.

expression has been found in the Kv1.1KO mouse brains (**Diez et al., 2003**). Carbamazepine, an anti-epileptic drug, could antagonize this increase of BDNF levels and also suppress the excessive neurogenesis in SGZ in adult Kv1.1KO mice (**Lavebratt et al., 2006**). We hypothesized that the excessive depolarization of type 2b neural progenitor cells lacking Kv1.1 (**Figure 4G** and **Figure 5G**) might stimulate neural progenitor cell proliferation by elevating the level of TrkB signaling. Restrained with the antibody incompatibility for double labeling, we stained the neural progenitor cells in SGZ with antibodies against phospho-TrkB (Tyr816), together with either Sox2 or Tbr2. We found most of the phospho-TrkB-positive cells within the SGZ region (**Figure 6**, **Figure 6—figure supplement 1**, and **Figure 7—figure supplement 1**), and a greater number of phospho-TrkB-positive cells in the dentate gyrus of Kv1.1KO mice as compared to the wild-type mice (**Figure 6**). Moreover, phospho-TrkB was found primarily in Tbr2+ or DCX+ cells (**Figure 6B** and **Figure 6—figure supplement 1**) but rarely in Sox2+ cells (**Figure 6A** and **Figure 6—figure supplement 1**), despite the fact that the expression levels of TrkB receptor are higher in type 1 neural progenitor cells (**Vilar and Mira, 2016**). Not only did the loss of Kv1.1 cause an increase of Tbr2+ cells with phospho-TrkB signals, but also the phospho-TrkB signals in the Tbr2+ cells were much more intense in the Kv1.1KO mice, suggesting that the TrkB signaling pathway was activated in the Tbr2-expressing type 2b cells in Kv1.1KO mice (**Figure 3**).

To test whether antagonizing TrkB activity could reduce the hyperplastic effect of Kv1.1 on adult neurogenesis, we performed daily intraperitoneal injections of GNF-5837 (20 mg/kg), a brain-permeable Trk inhibitor (**Albaugh et al., 2012**), into mice for 3 weeks starting at the first postnatal week. This 3-week GNF-5837 treatment was sufficient to suppress TrkB signaling, as the phospho-TrkB levels were drastically reduced in the hippocampus (**Figure 7—figure supplement 1**). Both Fezf2-GFP-positive cells and Ki67-positive cells in Kv1.1KO and control mice were comparable in number after this GNF-5837 treatment (**Figure 7B,C,D**). By contrast, age-matched Kv1.1KO mice receiving vehicle-only injection had significantly more Fezf2-GFP- and Ki67-positive cells in the SGZ than those in control mice (**Figure 7A,B,C**), in agreement with a previous report (**Almgren et al., 2007**). Thus, pharmacological inhibition of Trk receptors prevented the increase of neural progenitor cells in the SGZ of Kv1.1KO mice.

If Kv1.1 channels limit neural progenitor cell proliferation, Kv1.1KO neural progenitor cells should generate larger clones than controls. To estimate the clone size, we generated $Gli1^{creERT2/+}$; $Ntrk2^{flox/flox}$;$Rosa26^{Tom/+}$ and $Gli1^{creERT2/+}$;$Ntrk2^{flox/+}$;$Rosa26^{Tom/+}$ mice in the background of either wild-type or Kv1.1KO. These quadruple transgenic mouse lines carried a tamoxifen-inducible cre-recombinase $Gli1^{creERT2}$ that can sparsely delete the floxed-$Ntrk2$ gene that encodes the TrkB receptor in the neural stem cells and progenitor cells of the SGZ in the presence of tamoxifen. The Cre-dependent TdTomato reporter ($Rosa26^{Tom}$) was used to label tamoxifen-activated CreER recombinase cells and their progeny (**Li et al., 2013**; **Singh et al., 2015**). We intraperitoneally injected 3-week-old mice with a single dose of tamoxifen (0.5 mg/kg) to delete the floxed-$Ntrk2$ in the SGZ, and the clone sizes were estimated at 8 weeks of age. A single clone was defined as encompassing all the cells within a 100-μm radius of the clone center (**Singh et al., 2015**). Indeed, TrkB deletion abolished the hyperplastic phenotype in Kv1.1KO mice, as $Kcna1^{-/-}$;$Gli1^{creERT2/+}$;$Ntrk2^{flox/+}$;$Rosa26^{Tom/+}$ mice had larger clone sizes than $Gli1^{creERT2/+}$;$Ntrk2^{flox/+}$;$Rosa26^{Tom/+}$, $Kcna1^{-/-}$;$Gli1^{creERT2/+}$;

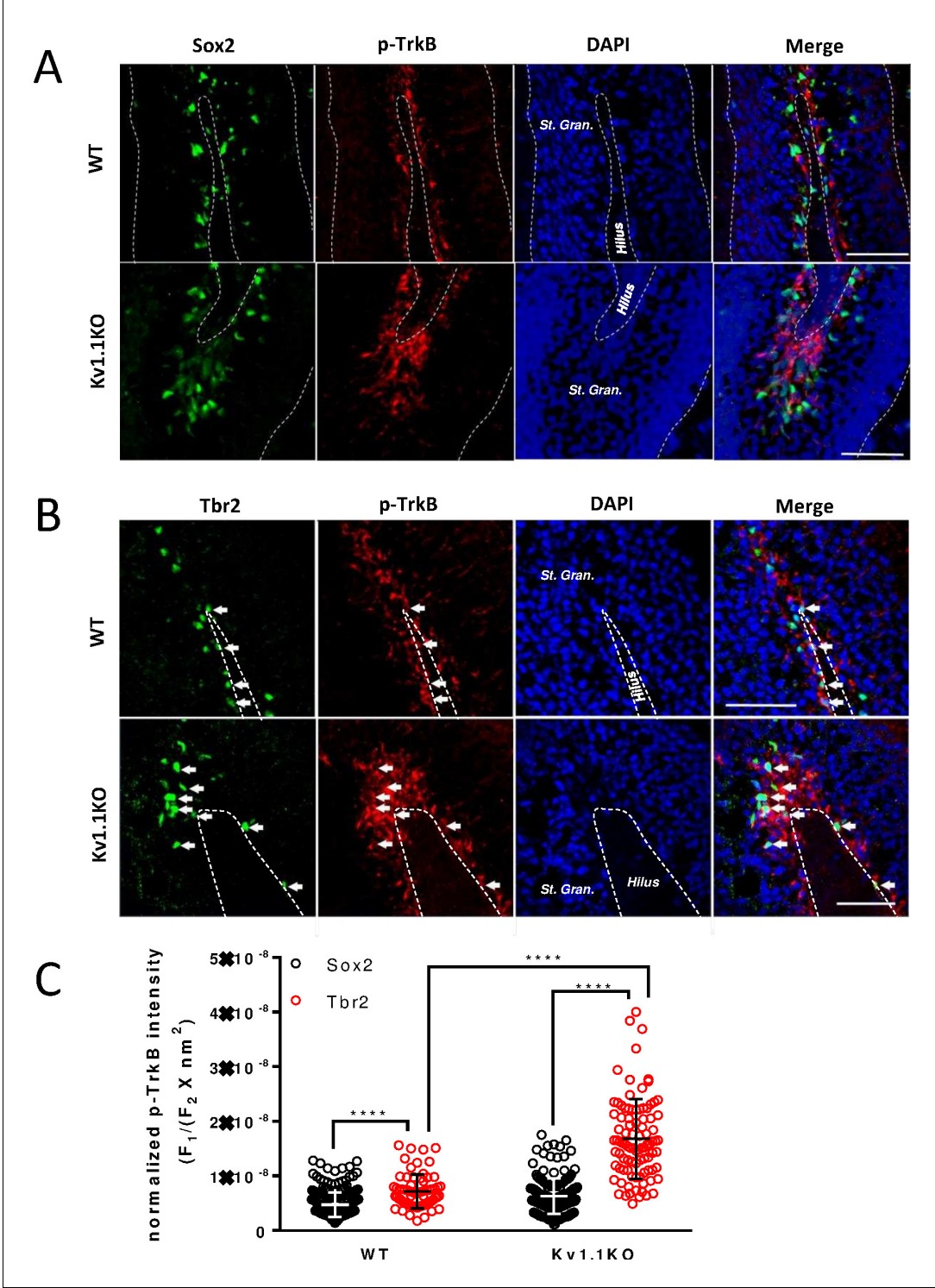

**Figure 6.** TrkB is mostly active in Tbr2-positive neural progenitor cells. Phospho-TrkB (Tyr816) signal represents a surrogate measure for TrkB activity. Phospho-TrkB-positive cells rarely colocalized with the Sox2-positive, presumably type 1 and type 2b neural progenitor cells (**A**); by contrast, phospho-TrkB-positive cells colocalized with the majority of Tbr2-positive, presumably type 2b cells. (**B**) More Tbr2+/phospho-TrkB+ neural progenitor cells could be found in the sub-granular zone (SGZ) of Kv1.1KO mice. Scale bar = 100 μm (**A**) and 50 μm (**B**). (**C**) Quantification analysis of phospho-TrkB levels in Sox2+ or Tbr2+ cells. The phospho-TrkB fluorescent intensity ($F_1$) of each Sox2+ or Tbr2+ cell was measured and normalized to the Sox2+ or Tbr2+ fluorescent intensity and

*Figure 6 continued on next page*

*Figure 6 continued*

the total measured cell area ($F_2$ X $nm^2$). n = 246, 243, 69, and 92 for Sox2+/WT, Sox2+/Kv1.1KO, Tbr2+/WT, and Tbr2+/Kv1.1KO cells, respectively. **** p<0.0001.

The online version of this article includes the following figure supplement(s) for figure 6:

**Figure supplement 1.** The TrkB signaling is active in DCX+ late-stage progenitor cells but not Sox2+ early-stage progenitor cells.

**Figure supplement 2.** Intraperitoneal injection of GNF-5837 remarkably inhibits the TrkB signaling in SGZ.

---

*Ntrk2*<sup>flox/flox</sup>;*Rosa26*<sup>Tom/+</sup>, and *Gli1*<sup>creERT2/+</sup>;*Ntrk2*<sup>flox/flox</sup>;*Rosa26*<sup>Tom/+</sup> mice (*Figure 7E* and *Figure 7—figure supplement 2*). The cumulative clone-size plots further confirmed a rightward skewness distribution rather than just an outlier bias (red trace) in *Kcna1*<sup>-/-</sup>;*Gli1*<sup>creERT2/+</sup>;*Ntrk2*<sup>flox/+</sup>;*Rosa26*<sup>Tom/+</sup> mice, and this shift was not observed for the neural progenitor cells lacking TrkB receptors (green trace) (*Figure 7F*).

## Discussion

In this study, we have shown that Kv1.1, a voltage-gated potassium channel, functions as a brake to fine-tune the rate of neurogenesis in mouse dentate gyrus within the first 2 months of postnatal life. Removing the Kv1.1 channel depolarized the late-stage neural progenitor cells that exhibit high input resistance and small capacitance (*Figure 5*), presumably the transit-amplifying type 2b neural progenitor cells (*Figure 4D*). Depolarization of type 2b neural progenitor cells causes cell-autonomous over-activation of the TrkB signaling pathway that promotes neural progenitor cell proliferation. Consequently, mice lacking functional Kv1.1 develop a megencephalic phenotype at early postnatal stages, as the neural progenitor cells without Kv1.1 over-proliferate due to elevated TrkB signaling (*Figure 8*).

In the central nervous system, mature neurons are capable of generating action potentials, while neural progenitor cells and immature neurons are considered electrically non-excitable and do not fire action potentials (*Bean, 2007*; *Fukuda et al., 2003*). In this study, our electrophysiological characterization of postnatal neural progenitors raised the possibility that alterations in membrane potential could impact postnatal neurogenesis. Although neural progenitor cells are not electrically excitable, these cells do express a variety of ion channels (*Shin et al., 2015*; *Yamashita, 2012*). The radial glia-like type 1 neural stem cells express high levels of glial inwardly rectifying potassium channels, mostly Kir4.1 and Kir5.1 (*Yasuda et al., 2008*). In addition to the inwardly rectifying potassium channels, type 1 neural stem cells in the SGZ are electrically coupled via gap junctions formed by connexins (*Figure 4*; *Kunze et al., 2009*; *Rozental et al., 1998*). Both inwardly rectifying potassium channels and gap junctions contribute to the extremely low input resistance in type 1 cells (*Figure 5E,H*) and keep these neural stem cells hyperpolarized (*Figures 4G* and *5D,G*). Once a type 1 cell commits to the neural progenitor cell fate, it will leave the extensive network of type 1 radial glia-like neural stem cells that are electrically coupled via gap junctions, as evidenced by the reduced capacitance and increased input resistance. Eventually, type 2b neural progenitor cells lose all gap junction connections and exist as single cells. Our multinomial logistic regression models faithfully depicted this process; when cells transited from the hyperpolarized network of quiescent type 1 neural stem toward individual highly proliferative type 2b neural progenitor cells, the input resistance was gradually increased, and the membrane capacitance was decreased due to the loss of gap junction coupling (*Figure 5E,F,H,I*). Moreover, our regression model provides a simple method for studying neural progenitor cell physiology. We showed that the identity of the neural progenitor cells within the SGZ could be accurately predicted based on the membrane potential, input resistance, and membrane capacitance obtained from patch-clamp recording, and there was no further need for post-hoc immunostaining of cell-type markers, a time-consuming procedure with a low success rate.

Among all voltage-gated potassium channels, the Kv1 family is crucial for regulating the resting membrane potential, as Kv1 channels can be activated at subthreshold membrane potentials (*Dodson et al., 2002*; *Grissmer et al., 1994*). Although Kv1.1 is expressed in most cells in the dentate gyrus, especially the inhibitory interneurons (yellow arrows, *Figure 2C*; *Li et al., 2012*), we only observed significant depolarization of type 2b neural progenitor cells lacking Kv1.1 (*Figures 4G* and

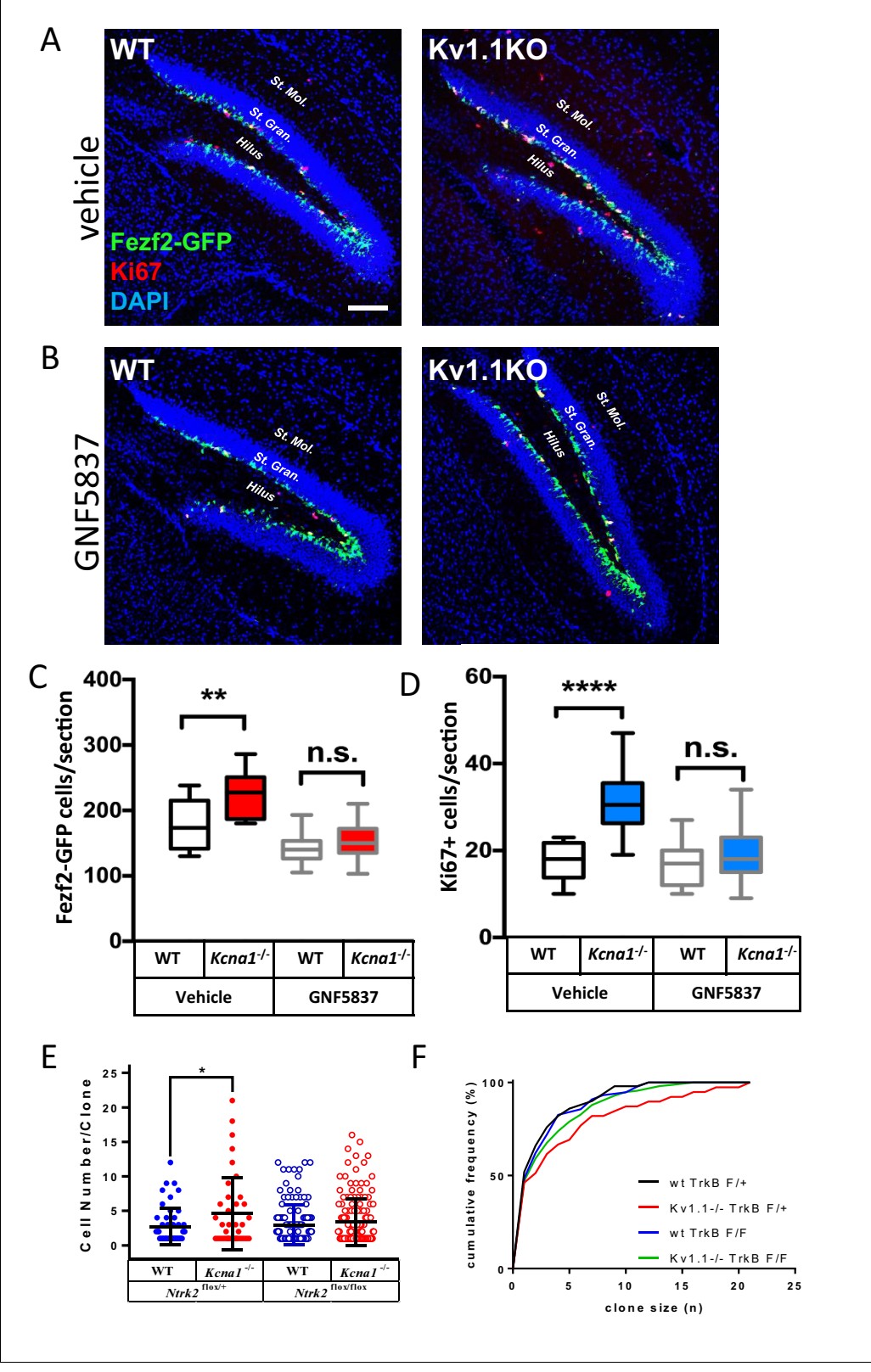

**Figure 7.** Kv1.1 channels regulate neural progenitor cell numbers via the TrkB signaling pathway. (**A, C, and D**) Kv1.1KO mice have more Fezf2-GFP- (**C**) and Ki67-positive cells (**D**) than wild-type mice, with vehicle treatment. p=0.0014, one-way ANOVA followed by Tukey's test in (**C**); p<0.0001, one-way ANOVA followed by Tukey's test in (**D**). (**B, C, and D**) Suppression of Kv1.1-dependent neurogenesis by GNF-5837, a potent Trk inhibitor. Mice

*Figure 7 continued on next page*

*Figure 7 continued*

receiving daily GNF-5837 (20 mg/kg) administration had comparable numbers of Fezf2-GFP- (C) and Ki67-positive cells (D) between wild-type and Kv1.1KO mice; n = 6–8 mice from each genotype with vehicle-only or GNF-5837 treatment. (E) Clonal analysis of adult neurogenesis. Tamoxifen (0.5 mg) was administered at postnatal 3 weeks to sparsely delete *Ntrk2* (a gene that encodes the TrkB receptor) in a subset of neural progenitor cells. In cells lacking only one allele of TrkB, Kv1.1KO had larger clone sizes than wild-type cells; by contrast, the Kv1.1-dependent proliferation advantage was abrogated in clones lacking both TrkB alleles. p=0.0389, comparing wild-type and Kv1.1KO on the *Ntrk2*$^{flox/+}$ background; two-way ANOVA followed by Sidak's multiple comparisons test; n = 39–157 for each group. Scale bar = 100 μm. Data are presented as mean ± SEM. *p<0.05, **p<0.01; ***p<0.001; ****p<0.0001; n.s., no significant difference. (F) Kolmogorov-Smirnov plots showing a rightward shift of clone sizes in *Kcna1*$^{-/-}$;*Gli1*$^{creERT2/+}$;*Ntrk2*$^{flox/+}$;*Rosa26*$^{Tom/+}$ mice (red trace) compared to wild-type *Gli1*$^{creERT2/+}$;*Ntrk2*$^{flox/+}$; *Rosa26*$^{Tom/+}$ mice (black trace) (p=0.0029, *t*-test), but this effect was absent in mice lacking TrkB receptor (p>0.05 between wild-type *Gli1*$^{creERT2/+}$;*Ntrk2*$^{flox/+}$;*Rosa26*$^{Tom/+}$ (black trace) and *Gli1*$^{creERT2/+}$;*Ntrk2*$^{flox/flox}$;*Rosa26*$^{Tom/+}$ (blue trace); *Kcna1*$^{-/-}$;*Gli1*$^{creERT2/+}$;*Ntrk2*$^{flox/flox}$;*Rosa26*$^{Tom/+}$ (green trace)).

The online version of this article includes the following figure supplement(s) for figure 7:

**Figure supplement 1.** Intraperitoneal injection of GNF-5837 remarkably inhibits the TrkB signaling in SGZ.
**Figure supplement 2.** Clonal analysis of postnatal born neurons in the SGZ.

---

*5D,G*). One plausible explanation for this specificity of effect is that the type 1 stem cells and type 2a neural progenitor cells remain hyperpolarized by the immense inwardly rectifying potassium currents and are further stabilized by the extensive electrical coupling through gap junctions. As stipulated by the Goldman-Hodgkin-Katz equation (*Hille, 2001*), conditions of low input resistance and large capacitance (*Figure 5E,F,H,I*) will diminish the effect of voltage-gated channels, such as Kv1.1, on the membrane potential to negligible levels. By contrast, a small potassium conductance could

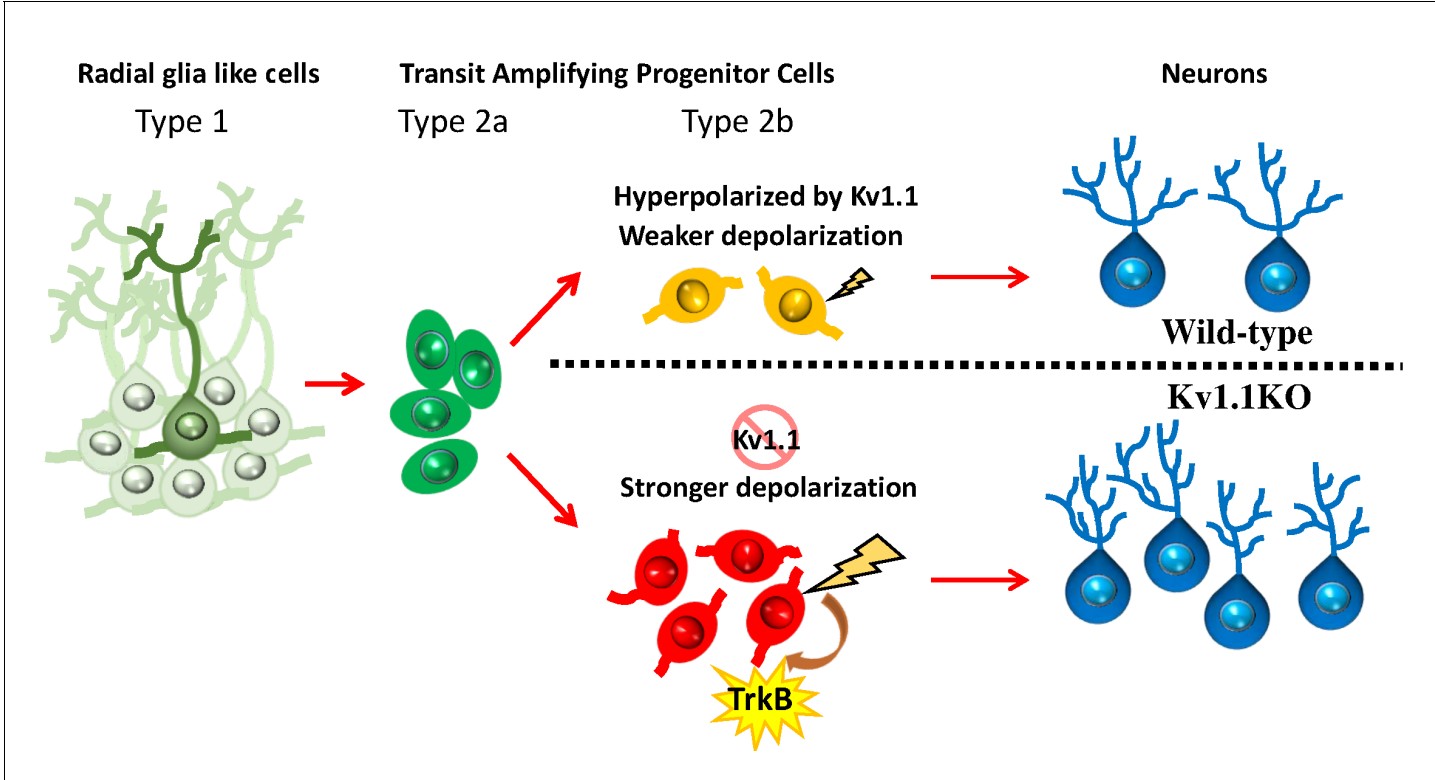

**Figure 8.** Summary of the role of Kv1.1 channels in regulating early postnatal neurogenesis in SGZ. The membrane potentials in type 1 radial glia-like neural stem cells and type 2a transit-amplifying progenitor cells are relatively hyperpolarized. In type 2b neural progenitors, cells lacking the Kv1.1 channel are more depolarized than wild types, further stimulating the proliferation of type 2b cells via activating the TrkB signaling pathway. The mature granule cells become hyperpolarized again.

be sufficient to keep cells with higher input resistance and smaller capacitance at relatively hyperpolarized membrane potentials.

Despite decades of extensive studies on postnatal neurogenesis, how ion channels and membrane potentials affect neurogenesis remains an intriguing open question. Ion channels have been shown to regulate postnatal and adult neurogenesis via both environmental (non-cell-autonomous) and intrinsic (cell-autonomous) mechanisms. Several studies have found that neural circuits may regulate the neural stem cell fate via non-cell-autonomous synaptic mechanisms. For example, increased gamma aminobutyric acid (GABA) tone of hippocampal parvalbumin-expressing interneurons facilitates the transition of radial glia-like cells into a quiescent state via γ2-containing GABA$_A$ receptors expressed in SGZ neural stem cells (*Song et al., 2012*). Furthermore, moderate activation of indirect glutamatergic mossy cells increases the radial glia-like stem cell quiescence (*Yeh et al., 2018*). On the other hand, several voltage-gated ion channels regulate neurogenesis in a cell-autonomous manner. Deleting *Paralytic*, the only voltage-gated sodium channel in *Drosophila*, shrinks the brain lobes due to reduced proliferation and enhanced apoptosis in type I and type II neuroblasts (*Piggott et al., 2019*). Moreover, in addition to regulating progenitor cell proliferation, hyperpolarization of progenitor cells in the cortex by inwardly rectifying potassium channels shifts neurogenesis from direct corticogenesis to indirect corticogenesis via suppression of WNT (Wingless-related integration site) activity (*Vitali et al., 2018*). In our study, the affected type 2b neural progenitor cells had already lost gap junctions and displayed high input resistance, so the minuscule Kv1.1 window current may be sufficient to keep wild-type but not Kv1.1KO type 2b neural progenitor cells hyperpolarized (*Figure 4G* and *Figure 5*; *Maylie et al., 2002*). This finding further raises the possibility that the plasticity of neurogenesis can be affected by bioelectric membrane properties (*Sundelacruz et al., 2009*).

One intriguing question concerns how membrane depolarization might activate the TrkB receptor in a cell-autonomous manner. The canonical ligands that activate the TrkB receptor are BDNF, released via the regulated secretory pathway, and neurotrophin 4/5 (NT4/5), released via the constitutive secretory pathway (*Bothwell, 2014*). BDNF is a plausible candidate for the endogenous neurotrophin that mediates the Kv1.1-dependent postnatal neurogenesis in mouse hippocampus, because BDNF is the only neurotrophin that has been detected in the SGZ in the mouse brain, and genetic deletion of NT4/5 did not impair adult neurogenesis (*Vilar and Mira, 2016*). BDNF plays a multifaceted role in the central nervous system, including promoting synaptogenesis, axonal guidance, dendrite outgrowth, and postnatal neurogenesis (*Jan and Jan, 2010*; *Jeanneteau et al., 2010*; *Liu and Nusslock, 2018*). Depolarizing the neural progenitor cells may elevate the intracellular calcium level through the voltage-gated calcium channels, which could trigger BDNF release via activating the SNARE complex (*Shimojo et al., 2015*; *Wong et al., 2015*; *Xu et al., 2018*). Furthermore, the unique biochemical properties of BDNF may allow such a diffusible signaling molecule to act cell-autonomously in regulating neurogenesis in mouse hippocampus. BDNF is a sticky peptide with multiple positively charged residues, and therefore, BDNF has a limited range of diffusion upon release (*Sasi et al., 2017*). In brain slices, BDNF only acts within 4.5 μm of its release site, which is less than the diameter of a single neural progenitor cell (*Mändl et al., 2002*). Hence, depolarization of the neural progenitor cells could release BDNF to activate TrkB signaling in a cell-autonomous manner due to its short effective range. Interestingly, GABA, another diffusible factor, has also been shown to cell-autonomously regulate neurogenesis in adult hippocampus. GABA activates the inhibitory GABA$_A$ receptors on the same neural progenitor cells by involving the GABA transporter VGAT. Removal of this inhibitory GABA signaling promotes neural progenitor cell proliferation by shortening the cell cycle via activation of S-phase checkpoint kinases and the histone variant H2AX (*Andäng et al., 2008*). Moreover, deleting a subset of GABA$_A$ receptors in neural progenitor cells not only stimulates neurogenesis, but also alters the positioning of newborn neurons in adult hippocampus, reminiscent of our finding that Kv1.1KO neurons are positioned farther away from the SGZ (*Figure 1F*; *Duveau et al., 2011*). Consistent with our results, over-activation of BDNF modulates the positioning in the SGZ of granule cells generated during early postnatal stages (*Kerloch et al., 2019*; *Scharfman et al., 2005*). Nevertheless, we cannot exclude the possible involvement of other signaling pathways, such as GSK3β and DISC1, since disrupting these signaling pathways causes failures of guidance for newborn neuron migration and positioning (*Duan et al., 2007*; *Ng et al., 2016*).

Given that previous studies have shown that TrkB receptors can be activated by mechanisms other than the neurotrophins, the elevated TrkB signaling we observed in Kv1.1KO mice could be BDNF-independent. For example, membrane depolarization itself could activate TrkB via elevating the intracellular cAMP levels, which has been demonstrated in cultured neurons previously (*Meyer-Franke et al., 1998*). Under certain circumstances, TrkB receptors can also be auto-activated in the absence of neurotrophins by increasing the receptor abundance (*Gupta et al., 2020*). The immature form of TrkB receptor that lacks N-glycosylation on the extracellular domain is constitutively active, even if it is located intracellularly; the membrane potential could indirectly regulate TrkB activity by tuning the cellular glycosylation process (*Gupta et al., 2020*; *Watson et al., 1999*). Moreover, TrkB receptors can also be transactivated via cross-talk with certain G-protein-coupled receptors, such as EGF (*Puehringer et al., 2013*) and adenosine A2a receptors (*Lee and Chao, 2001*). Lastly, TrkB signaling can be elevated by increasing *Ntrk2* gene expression (*Yamashita et al., 2021*). It will be of interest to pursue experiments to investigate whether these mechanisms are involved in Kv1.1-dependent postnatal neurogenesis in mouse hippocampus.

One important limitation of our study is the age of animals we used. To avoid the confounding effects of seizures that begin in 3- to 4-week-old Kv1.1KO mice, we focused our study on early postnatal neurogenesis. A recent work using virally labeled neural progenitor cells in the SGZ has revealed a massive burst of postnatal neurogenesis within the first week, as evidenced by viral labeling of neural progenitor cells at P6; these neurons persist for about 2 months before undergoing apoptosis, a time course distinct from adult-born neurons produced later in life (*Ciric et al., 2019*). Our study of young mice describes the involvement of Kv1.1 in the generation of neurons by neural progenitor cells beginning in the first postnatal week. How the loss of Kv1.1 function affects neurogenesis later in life will require further study.

In summary, our study has uncovered a unique role of Kv1.1 in regulating neurogenesis within the postnatal dentate gyrus during the first 2 months of life. Alterations in adult neurogenesis in the dentate gyrus have been reported for patients with psychiatric and neurological diseases (*Gonçalves et al., 2016*; *Sahay and Hen, 2007*), and recent studies have suggested that neural progenitor cells might be potential targets for treating neurodegenerative disorders such as Alzheimer's disease and Parkinson's disease (*Christie and Turnley, 2012*). Thus, our findings regarding Kv1.1 modulation of early postnatal neurogenesis may help lay the groundwork for future identification of novel therapeutic targets for neurodegenerative and psychiatric diseases (*Christian et al., 2014*).

# Materials and methods

## Key resources table

| Reagent type (species) or resource | Designation | Source or reference | Identifiers | Additional information |
|---|---|---|---|---|
| Strain background (*Mus musculus*) | ICR (CD-1 or Swiss Outbred) | BioLASCO Taiwan | | Outbred strain |
| Genetic reagent (*Mus musculus*) | *Kcna1*[-] (*Kcna1*[tm1Tem]) | PMID:9581771 | MGI: 1861959 | Dr. Bruce Tempel (University of Washington, USA) |
| Genetic reagent (*Mus musculus*) | *Rosa26*[GT] (Gt(*ROSA*)26[Sortm6(ACTB-EGFP*,-tdTomato)]) | PMID:22479386 | MGI: 5314252 | Dr. Liqun Luo (Stanford University, USA) |
| Genetic reagent (*Mus musculus*) | *Rosa26*[TG] (*Gt(ROSA)26*[Sortm7(ACTB-EGFP*)]) | PMID:22479386 | MGI: 5314253 | Dr. Liqun Luo (Stanford University, USA) |
| Genetic reagent (*Mus musculus*) | POMC-GFP (Tg(Pomc-EGFP)1) | PMID:11373681 | MGI: 3776091 | Dr. Jeffery Friedman (Rockefeller University, USA) |
| Genetic reagent (*Mus musculus*) | Fezf2-GFP (Tg(Fezf2-EGFP)CO61Gsat) | PMID:14586460 | MGI: 3847288 | Dr. Su Guo (University of California, San Francisco, USA) |
| Genetic reagent (*Mus musculus*) | *Ntrk2*[flox] (*Ntrk2*[tm1Lfr]) | PMID:10995833 | MGI: 2384391 | Dr. Louis Reichardt (University of California, San Francisco, USA) |
| Genetic reagent (*Mus musculus*) | Nestin-cre (Tg(Nes-cre)1Kln) | PMID:10471508 | MGI: 2176173 | Jackson Laboratory, USA |

*Continued on next page*

*Continued*

| Reagent type (species) or resource | Designation | Source or reference | Identifiers | Additional information |
|---|---|---|---|---|
| Genetic reagent (*Mus musculus*) | *Gli1*^creERT2 (*Gli1*^tm3(cre/ERT2)Alj) | PMID:15315762 | MGI: 3053957 | Jackson Laboratory, USA |
| Genetic reagent (*Mus musculus*) | *Rosa26*^Tom (Gt(*ROSA*)26^Sortm14(CAG-tdTomato)Hze) | PMID:20023653 | MGI: 3809524 | Jackson Laboratory, USA |
| Antibody | Anti-GFP (chicken polyclonal) | Aves, USA | Cat#: GFP-1010 RRID:AB_2307313 | 1:400 |
| Antibody | Anti-Sox2 (mouse monoclonal, IgG2b) | Millipore, USA | Cat#: MAB4343 RRID:AB_11205572 | 1:400 |
| Antibody | Anti-Tbr2 (rabbit monoclonal) | Abcam, USA | Cat#: ab183991 RRID:AB_2721040 | 1:400 |
| Antibody | Anti-Tbr2 (chicken polyclonal) | Millipore, USA | Cat#: AB15894 RRID:AB_10615604 | 1:400 |
| Antibody | Anti-Doublecortin (rabbit monoclonal) | Cell Signaling, USA | Cat#: 14802 | 1:400 |
| Antibody | Anti-phospho-TrkB (Tyr816) (rabbit polyclonal) | Millipore, USA | Cat#: ABN1381 RRID:AB_2721199 | 1:100 |
| Antibody | Anti-Kv1.1 (mouse polyclonal, IgG2b) | Aves, USA | Cat#: 75–105RRID:AB_2128566 | 1:200 |
| Antibody | Anti-Ki67 (rabbit polyclonal) | Invitrogen, USA | Cat#: MA5-14520 RRID:AB_10979488 | 1:500 |
| Commercial assay or kit | Fluoromount G | SouthernBiotech, USA | Cat#: 0100–01 RRID:SCR_015961 | |
| Commercial assay or kit | RNAscope Fluorescent Multiplex Kit | Advanced Cell Diagnostics, Inc, USA | Cat#: 320850 | |
| Recombinant DNA reagent | Probe-Mm-GFP | Advanced Cell Diagnostics, Inc, USA | Cat#: 400281 | |
| Recombinant DNA reagent | Probe-Mm-Kcna | Advanced Cell Diagnostics, Inc, USA | Cat#: 435901 | |
| Chemical compound, drug | GNF-5837 | Tocris, UK | Cat#: 4559 | |
| Chemical compound, drug | PEG400 (Polyethylene glycol 400) | Sigma-Aldrich, USA | Cat#: 1546445 | |
| Chemical compound, drug | Cremophor EL | Sigma-Aldrich, USA | Cat#: C5135 | |
| Chemical compound, drug | Dendrotoxin-K | Alomone Labs, Israel | Cat#: D-400 | |
| Chemical compound, drug | Cremophor EL | Sigma-Aldrich, USA | Cat#: C5135 | |
| Chemical compound, drug | Neurobiotin | Vector Laboratories, USA | Cat#: SP-1120 | 0.3% in the pipette solution |
| Chemical compound, drug | Lucifer yellow-conjugated dextrans | Invitrogen, USA | Cat#: D-1825 | 0.2% in the pipette solution |
| Chemical compound, drug | Neutravidin | Thermo Fisher, USA | Cat#: 31000 | 0.25% |
| Software, algorithm | pClamp 10.0 | Molecular Devices, USA | RRID:SCR_011323 | |
| Software, algorithm | GraphPad Prism 7.0 | Graphpad, USA | RRID:SCR_002798 | |
| Software, algorithm | Imaris 9.0 | Bitplane, Switzerland | RRID:SCR_007370 | |
| Software, algorithm | R Project for Statistical Computing | The R foundation | RRID:SCR_001905 | |

## Animals

This study was carried out in strict accordance with the recommendations found in the Guide for the Care and Use of Laboratory Animals of the National Institutes of Health. The experimental protocols were approved by the Institutional Animal Care and Use Committee of Academia Sinica (protocol #: 15-01-813) and the University of California, San Francisco. Mice (3–5 per cage) were housed in the animal facility and fed with a regular chow diet on a standard 12 hr light/12 hr dark cycle. At least three animals were used for every experimental group. The $Kcna1^-$ mice were obtained from Dr. Bruce Tempel's lab at the University of Washington, USA. $Rosa26^{GT}$ (Gt(ROSA)26$^{Sortm6(ACTB-EGFP*,-tdTomato)}$) and $Rosa26^{TG}$ (Gt(ROSA)26$^{Sortm7(ACTB-EGFP*)}$), two transgenic mouse lines for producing MADM-6 mice, were obtained from Dr. Liqun Luo's lab at Stanford University, USA. The $Ntrk2^{flox}$ mice were obtained from Dr. Louis Reichardt's lab at the University of California, San Francisco, USA. POMC-GFP (Tg(Pomc-EGFP)1) mice were obtained from Dr. Jeffery Friedman's lab at Rockefeller University, USA. Fezf2-GFP (Tg(Fezf2-EGFP)CO61Gsat) mice were obtained from Dr. Su Guo's lab at the University of California, San Francisco, USA. Nestin-cre (Tg(Nes-cre)1Kln), $Gli1^{creERT2}$ ($Gli1^{tm3(cre/ERT2)}$), and $Rosa26^{Tom}$ (Gt(ROSA)26$^{Sortm14(CAG-tdTomato)}$) mice were obtained from Jackson Laboratory (Bar Harbor, Maine, USA). All mice were maintained on an ICR (Institute of Cancer Research) background. GNF-5837 (Tocris, UK) was dissolved in dimethyl sulfoxide (DMSO) (Sigma, USA), and then diluted in a solvent composed of 65% PEG400 (Sigma, USA) and 35% Cremophor EL (Sigma, USA) for intraperitoneal injection.

## Electrophysiology

Brain slices (250 µm thickness) containing the hippocampus were prepared as described previously (*Yang et al., 2012b*). Mice were first anesthetized with isoflurane and then decapitated. The brain was swiftly removed and placed in cutting solution: 110 mM choline chloride, 25 mM $NaHCO_3$, 11 mM glucose, 2.5 mM KCl, 1.25 mM $NaH_2PO_4$, 11.6 mM sodium ascorbate, 3.1 mM sodium pyruvate, 7 mM $MgCl_2$, and 0.5 mM $CaCl_2$, equilibrated with 95% $O_2$/5% $CO_2$. Brain slices were obtained from tissue immersed in the cutting solution using a compresstome (Precisionary Instruments, USA). Slices were then incubated in artificial cerebrospinal fluid (aCSF): 126 mM NaCl, 21.4 mM $NaHCO_3$, 10 mM glucose, 2.5 mM KCl, 1.25 mM $NaH_2PO_4$, 1.2 mM $MgCl_2$, and 2 mM $CaCl_2$, equilibrated with 95% $O_2$/5% $CO_2$. An Axon700B amplifier (Molecular Devices Corp, USA) was used to measure membrane currents and membrane capacitance in the standard whole-cell patch-clamp configuration. Data were acquired at 5 kHz with Clampex10 software (Molecular Devices Corp, USA). The intracellular solution contained 135 mM potassium gluconate, 15 mM KCl, 10 mM HEPES, 5 mM $Mg_2ATP$, 1 mM $Na_3GTP$, 10 mM sodium phosphocreatine, and 0.05 mM EGTA; pH was adjusted to 7.2 with KOH. Dendrotoxin-κ (Alomone Labs, Israel) was used to specifically block the Kv1.1 channel. Pipettes were pulled from 1.5 mm borosilicate glass capillaries (Sutter Inc, USA) and had a resistance of 3–5 MΩ when filled with the intracellular solution; data were excluded from further analyses if the series resistance was higher than 20 MΩ and the holding current at −70 mV was lower than −100 pA. All experiments were performed at room temperature.

## Immunostaining

Mice were fed ad libitum and were anesthetized with intraperitoneal Zoletil/Xylazine injection before transcardial perfusion with saline followed by 4% PFA. Brains were removed and post-fixed overnight in 4% PFA. Brains were then cryoprotected overnight in saline containing 30% sucrose at 4°C until they sank. For antigen retrieval, the brain sections (16 µm) were boiled in sodium citrate buffer (10 mM sodium citrate, 0.05% Tween 20, pH 6.0) between 95 and 100°C for 20 min. After the buffer was cooled to room temperature, the samples were further processed. The brain sections were washed in a blocking medium containing 0.1% Triton X-100% and 5% donkey serum (Jackson Immunoresearch Laboratories), and incubated overnight (4°C) with primary antibodies against GFP (chicken, 1:400; Aves, USA), Sox2 (mouse IgG2b, 1:400; Millipore, USA), Ctip2 (rabbit, 1:500; Abcam, USA), Tbr2 (rabbit, 1:400; Abcam, USA.), Tbr2 (chicken, 1:400; Millipore, USA), Doublecortin (rabbit, 1:400; Cell Signaling, USA), Kv1.1 (1:200; Antibodies Inc, USA), phospho-TrkB (rabbit, 1:200; Millipore, USA), or Ki67 (rabbit, 1:500; Invitrogen, USA), followed by Alexa dye-tagged secondary antibodies (donkey 1:100; Invitrogen, USA). For immunostaining of samples after electrophysiology recording, the brain slices were fixed in 4% PFA at 4°C for 2 hr. The internal solution contained

Neurobiotin (0.3%, MW: 367 Da; Vector Laboratories, USA) and Lucifer yellow-conjugated dextrans (0.2%, MW: 10 kDa; Invitrogen, USA) to respectively label gap junction-coupled cells and recorded cells. The slices were washed three times at room temperature for 40 min in 0.3% Triton X-100 and 3% Bovine Serum Albumin (BSA) (Sigma Aldrich, USA) for blocking and permeabilization, followed by incubation overnight at 4°C with Neutravidin (0.25%; Thermo Fisher Scientific, USA) and primary antibodies against GFP (chicken, 1:400; Aves, USA), Sox2 (mouse IgG2b, 1:400; Millipore, USA) or Tbr2 (rabbit, 1:400; Abcam, USA). Secondary antibodies conjugated with goat anti-rabbit IgG Alexa 405, goat anti-chicken IgG Alexa 488, and goat anti-mouse IgG2b Alexa 633 were purchased from Invitrogen. The slides were mounted using Fluoromount G mounting medium containing 4′,6-diami-dino-2-phenylindole (DAPI) (Southern Biotech, USA), and images were acquired using a confocal microscope (Zeiss, Germany).

## In situ hybridization

The mouse brains were frozen and sectioned, as previously described (*Li et al., 2019*). Sections were fixed with chilled 4% PFA for 15 min and then washed with 0.1 M phosphate buffered saline (PBS) twice. Sections were then dehydrated with 50% ethanol, 70% ethanol, and 100% ethanol sequentially. Slides were treated with protease and then incubated with a customized probe for 2 hr. Signal was detected using Probe-Mm-GFP and Probe-Mm-Kcna and further amplified by RNAscope Fluorescent Multiplex Kit (Advanced Cell Diagnostics, USA). Images were acquired using a confocal microscope (Zeiss, Germany).

## Clonal analysis

The clonal analysis was performed as described previously (*Singh et al., 2015*). Briefly, for sparsely labeling the progenitor cells, 3-week-old mice carrying $Gli1^{creERT2}$; $Rosa26^{Tom}$ with wild-type or $Kcna1^{-/-}$ and $Ntrk2^{flox/flox}$ or $Ntrk2^{flox/+}$ were injected with a single dose of 0.5 mg/kg of tamoxifen (Sigma, USA) dissolved in corn oil (Sigma, USA). At 8 weeks of age, mice were killed by an intraperitoneal injection of 150 mg/kg Zoletil + 20 mg/kg xylazine. Both hippocampi were dissected out and fixed overnight in a PBS solution containing 4% PFA. The hippocampi were rendered transparent in scale 0 for 2 days at 37°C, and then transferred to scale 4 for 1 day at 4°C (*Hama et al., 2015*). The whole-mount hippocampus was imaged in 3D using a Zeiss LSM700 confocal microscope system (Zeiss, Germany). Imaris9.7 (Oxford Instruments, Switzerland) was used to identify TdTomato-positive neural progenitor cells. The clonal clusters were defined as the TdTomato-positive cells within 100-µm radius of the clone center, as illustrated in *Figure 7—figure supplement 1* (*Singh et al., 2015*).

## Multinomial logistic regression classification of cell types

To classify cells and make associations between cell type and resting membrane potentials, input resistance and membrane capacitance, we applied the multinomial logistic regression model as follows:

$$\frac{P(C=c|X=x)}{P(C=0|X=x)} = \exp\left(\beta_{c,0} + x_1\beta_{c,1} + x_2\beta_{c,2} + x_3\beta_{c,3}\right), c = 1, 2,$$

where C = 0, 1, 2 represents type 1, 2a, and 2b cells, respectively, and X = (X₁,X₂,X₃) represents the centralized and standardized $(\log Cm, \log Rm, Vm)$. By using a standard maximum likelihood estimation, the estimated coefficients are

$$\left(\beta_{1,0}, \beta_{1,1}, \beta_{1,2}, \beta_{1,3}, \beta_{2,0}, \beta_{2,1}, \beta_{2,2}, \beta_{2,3}\right) = (-0.343, -1.067, -1.317, 1.681, -3.074, -2.746, 0.782, 1.359)$$

for wild-type cells and

$$\left(\beta_{1,0}, \beta_{1,1}, \beta_{1,2}, \beta_{1,3}, \beta_{2,0}, \beta_{2,1}, \beta_{2,2}, \beta_{2,3}\right) = (0.567, -1.373, -0.051, 0.302, -5.353, -6.935, -1.626, 5.549)$$

for Kv1.1KO cells.

Each cell was then assigned to the class with the highest probability $P(C = c|X = x), c = 0, 1, 2$. Based on the estimated coefficients, it appeared that smaller values of Cm and larger values of Vm

were associated with a higher probability of being type 2b neural progenitor cells than being type 1 neural stem cells.

## Statistical analyses

For immunostaining experiments, cell counts were performed on two to three images per mouse, and $n$-values indicate numbers of different mice, except for the assessment of clonal analysis, where $n$-values correspond to numbers of different clonal pools from four to six mice per genotype. For electrophysiological experiments, $n$-values correspond to numbers of different cells from each individual brain slice. Statistical analyses were performed with Prism software (Graphpad, USA) or R Project for Statistical Computing (The R Foundation, USA). Two-way ANOVA with Sidak's multiple comparisons post-hoc test, one-way ANOVA with Tukey's test, Mann-Whitney $U$-test, or Student's $t$-test for pair-wise comparisons were used as appropriate. $p < 0.05$ was considered statistically significant.

## Acknowledgements

We thank Drs. Su Guo (University of California, San Francisco), Bruce Tempel (University of Washington), Liqun Luo (Stanford University), and Hui Zong (University of Oregon) for providing transgenic mice. We thank the outstanding technical supports provided by the Neuro-imaging core facility of the Neuroscience Program, Microscopy facility and Electrophysiology facility of the Institute of Biomedical Sciences, Academia Sinica. This work was supported by the Institute of Biomedical Sciences at Academia Sinica and the Ministry of Science and Technology (106–2320-B-001–013 and 107–2320-B-001–026-MY3 to SBY) and by the National Institute of Health (R01MH065334 to LYJ). YNJ and LYJ are Howard Hughes Medical Institute investigators.

## Additional information

### Competing interests

Grant Guangnan Li: Grant Guangnan Li is affiliated with Nkarta Therapeutics Inc, The author has no financial interests to declare.". The other authors declare that no competing interests exist.

### Funding

| Funder | Grant reference number | Author |
| --- | --- | --- |
| Ministry of Science and Technology, Taiwan | 106-2320-B-001-013 | Shi-Bing Yang |
| Ministry of Science and Technology, Taiwan | 107-2320-B-001-026-MY3 | Shi-Bing Yang |
| NIH Blueprint for Neuroscience Research | R01MH065334 | Lily Yeh Jan |
| Howard Hughes Medical Institute | | Yuh Nung Jan Lily Yeh Jan |

The funders had no role in study design, data collection and interpretation, or the decision to submit the work for publication.

### Author contributions

Shu-Min Chou, Data curation, Formal analysis, Investigation, Visualization, Methodology, Writing - review and editing; Ke-Xin Li, Data curation, Formal analysis, Validation, Investigation, Visualization, Methodology, Writing - review and editing; Ming-Yueh Huang, Data curation, Software, Formal analysis, Investigation, Visualization, Methodology, Writing - original draft, Writing - review and editing; Chao Chen, Data curation, Formal analysis, Visualization, Writing - review and editing; Yuan-Hung Lin King, Wei Zhou, Data curation, Formal analysis; Grant Guangnan Li, Resources, Investigation, Methodology; Chin Fen Teo, Data curation; Yuh Nung Jan, Resources, Supervision, Funding acquisition, Investigation; Lily Yeh Jan, Resources, Supervision, Funding acquisition, Investigation,

Methodology, Project administration, Writing - review and editing; Shi-Bing Yang, Conceptualization, Resources, Data curation, Formal analysis, Supervision, Funding acquisition, Validation, Investigation, Visualization, Methodology, Writing - original draft, Project administration, Writing - review and editing

### Author ORCIDs
Ke-Xin Li ⓘ http://orcid.org/0000-0003-3879-294X
Yuh Nung Jan ⓘ http://orcid.org/0000-0003-1367-6299
Lily Yeh Jan ⓘ https://orcid.org/0000-0003-3938-8498
Shi-Bing Yang ⓘ https://orcid.org/0000-0001-8061-3963

### Ethics
Animal experimentation: This study was carried out in strict accordance with the recommendations in the Guide for the Care and Use of Laboratory Animals of the National Institutes of Health, and used protocol approved by the Institutional Animal Care and Use Committee of Academia Sinica (protocol#:15-01-813) and the University of California, San Francisco. Mice (3-5 per cage) housed in the animal facility were fed with regular chow diet and subjected to a standard 12-h light/12-h dark cycle. At least 3 animals were used for every single experiment. Mice were first anesthetized with isoflurane followed by decapitation for electrophysiological recordings and immunostaining.

### Decision letter and Author response
Decision letter https://doi.org/10.7554/eLife.58779.sa1
Author response https://doi.org/10.7554/eLife.58779.sa2

## Additional files

### Supplementary files
• Transparent reporting form

### Data availability
Most of our results are presented as scatterplots with the intention to show the distribution of our raw data. The variables for the multinomial logistic regression model (fig 5) can be found in the methods section.

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
