## [Decision Letter]

**Acceptance summary:**

This manuscript investigates the mechanism by which Kv1.1 potassium channels regulate postnatal neurogenesis. This is an important topic since the physiological and molecular mechanisms that regulate the decision of neurogenesis versus differentiation are not well understood. In this study, a loss of Kv1.1 channels increased proliferation of progenitor cells, which is attributed to a depolarization of the membrane potential and signaling through the BDNF/TrkB receptor system.

**Decision letter after peer review:**

Thank you for submitting your article "Kv1.1 channels regulate early postnatal neurogenesis in mouse hippocampus via the BDNF-TrkB pathway" for consideration by *eLife*. Your article has been reviewed by 3 peer reviewers, and the evaluation has been overseen by a Reviewing Editor and Kenton Swartz as the Senior Editor. The following individuals involved in review of your submission have agreed to reveal their identity: Robert Blum (Reviewer #1); Juan P. Zanin (Reviewer #3).

The reviewers have discussed the reviews with one another and the Reviewing Editor has drafted this decision to help you prepare a revised submission.

Summary:

This manuscript addresses the mechanism through which Kv1.1 potassium channels regulate postnatal neurogenesis in the SGZ of the dentate gyrus. This is a significant topic because the physiological and molecular mechanisms regulating neurogenesis vs. differentiation are not well understood. Specifically, this paper investigates a presumed cell-autonomous increase in early postnatal neurogenesis in mice that lack Kv1.1. Deletion of the potassium channel shifts the resting membrane potential to a more depolarized state. There is evidence presented that loss of Kv1.1 increased proliferation of type 2b progenitor cells, and this increased proliferation is attributed to a depolarization of the membrane potential, leading to cell-autonomous BDNF/TrkB-mediated signaling.

Overall, this is a potentially exciting set of results but the main conclusions in the paper require the support of additional data, as detailed below.

Essential revisions:

The role of BDNF-TrkB signaling is not convincingly established. More evidence is needed to show BDNF-TrkB signaling is responsible for the described main effect (more cells in Kv1.1KO, an effect that was already described in 2012). The mechanism, namely that Kv1.1 KO increases excitability in Type 2b cells and causes TrkB activation to speed up cell division is plausible, but not shown. However, it would be possible to produce such data. It is essential that the link between Kv1.1-dependent depolarization and BDNF-TrkB induced proliferation be better supported.

Figure 6 attempted to provide evidence that BDNF/TrkB signaling is required for increased proliferation of Type 2b cells. However no support is provided of the selectivity of GNF5837. At best, it is stated to be a Trk inhibitor, not selective for TrkB. There is also no rationale or explanation of the use of Ki67 as a marker. For the experiments using TrkB deletion, the difference between WT and KO in the heterozygous deletion is not convincing because it appears to be accounted for by a few outliers, and there is no discussion of why this difference is not seen with homozygous TrkB deletion.

Experimental concerns

1. Figure 1E: It is not clear what the authors measured and how they did it. The y-axis tells us something about distance from SGZ in percent. The interpretation in direction of increased proliferation is not unlikely, especially in context of figure 7. However, S-phase shortening ihas been done at postnatal day 10, a time point when BDNF expression is rather low and under developmental control. The data are not linked and therefore not fully conclusive at this point of investigation.

2. Figure 2: In Figure 2, we expect to see that Kv1.1 is expressed in the dentate gyrus and is present in Fezf2-positve cells. The images are not easy to interpret. Overview images are missing. Where are these in situ dots? Which controls were used and how did the controls look like? I suppose that these are the progenitors close to the SVZ. Later in their study, Kv1.1 becomes functionally important in type2b cells or MADM-labelled cells within the DGin Fezf2-GFP positive cells. This image does not clarify whether Kv1.1 is also expressed in these cells, albeit it is very likely. Furthermore, immunolabeling of Kv1.1 minus mice (IHC) is missing. IHC, using this protocol, should be able to check immunoreactivity in the type2b cell type. This is important, as the authors provide evidence for changes in electrophysiology in type2b cells, but not the other cells that also express Kv1.1.

3. For Figure 2D-G, in order to support the conclusion that there are functional Kv1.1 channels in these progenitor cells, it is essential to present the raw un-subtracted current traces before and after Dtx-K application.

4. Figure 3C shows that there is a higher proportion of type 2b cells compared to other subtypes in the Kv1.1 knockout. This is interpreted as increased cell proliferation in the absence of Kv1.1. However, since Type 2A cells differentiate to Type 2b, a higher proportion of Type 2b cells could instead be accounted for by enhanced differentiation. This issue is not addressed.

5. Figure 3D shows that in Type 2b cells specifically, KV1.1 KO cells had a more depolarized RMP than WT. It is not clear if a sufficient number of cells was recorded for each group for this critical experiment, the figure legend reports a wide range of n=5 to 28 cells per group. The data in 3D are rather important, but are not well explained. Data should be presented as single data points. As the 2b cell type is in the focus of the analysis, a table with the cell type prediction data would help. The main effect is type 2b WT versus Kv1.1 KO, but a direct comparison of the data is missing. The authors should better explain why they used this statistical test (hypothesis?) and why they did the statistical comparison between the different groups.

6. It has not been shown whether the systemically applied Trk inhibitor affects Trk phosphorylation in the hippocampus, meaning across the blood brain barrier, in these experiments.

7. Clonal analysis in Figure 6A-C affects Fezf2-GFP, but the criteria for clonal analysis is not well described. By going through the methods I realized that this is the 3D analysis of clones using Td Tomato. Image data of this analysis should be presented and the way of analysis must be clear. As it is not easy to obtain unbiased analysis of clone sizes with such an approach, it is important to show how it has been done and how the images look like.

8. The most critical point is that knocking out TrkB from adult born neurons is sufficient to interfere with neurogenesis (e.g. Bergami et al. PNAS 2008) and this effect was not be connected to Kv1.1. So the links between BDNF, TrkB, and more division at early stages are not causally connected. The overall idea is that Type2b cells that lack Kv1.1 are briefly depolarized, that this causes local BDNF release and a stimulation of cell division. However, it has not been shown that these cells, or close neighbors express more BDNF, that this stimulates TrkB and that this specific cell type causes the increase in adult born neuron numbers. Furthermore, it may be that the increase in the RMP does not lead to a higher activity of the neuron and activity-dependent gene expression of BDNF, because an increase in this parameter may induce a depolarization block; unlikely, but not ruled out and a relevant problem in excitability research.

---

## [Author Response]

Essential revisions:The role of BDNF-TrkB signaling is not convincingly established. More evidence is needed to show BDNF-TrkB signaling is responsible for the described main effect (more cells in Kv1.1KO, an effect that was already described in 2012). The mechanism, namely that Kv1.1 KO increases excitability in Type 2b cells and causes TrkB activation to speed up cell division is plausible, but not shown. However, it would be possible to produce such data. It is essential that the link between Kv1.1-dependent depolarization and BDNF-TrkB induced proliferation be better supported.

To make the connection between Kv1.1KO and BDNF-TrkB signaling, we aimed to investigate whether the phospho-TrkB levels were elevated in Kv1.1KO type 2b cells. The first step of activation of TrkB by BDNF is the auto-phosphorylation of TrkB receptor at Tyr816, and therefore, we expected to detect phospho-TrkB signals in cells with BDNF-TrkB activation. As expected, we did observe elevated phospho-TrkB levels in the SGZ in Kv1.1KO mice (Figure 6 and Figure 6—figure supplement 1). Moreover, we noticed that the phospho-TrkB levels were much elevated in cells expressing Tbr2 or doublecortin but not Sox2, indicating that in Kv1.1KO mice, the BDNF-TrkB was preferentially activated in late-stage neural progenitor cells (Figure 6C). Both pharmacological inhibition and genetic deletion of Trk signaling prevented the increase of BDNF-TrkB activation in Kv1.1KO neural progenitor cells, as well as suppressing the over-production of neural progenitor cells and reducing the clone size (Figure7). These new results support the association of the hyperplastic effect of Kv1.1KO with the overactive BDNF-TrkB signaling primarily in late-stage neural progenitor cells that are depolarized in the absence of Kv1.1.

Figure 6 attempted to provide evidence that BDNF/TrkB signaling is required for increased proliferation of Type 2b cells. However no support is provided of the selectivity of GNF5837. At best, it is stated to be a Trk inhibitor, not selective for TrkB. There is also no rationale or explanation of the use of Ki67 as a marker. For the experiments using TrkB deletion, the difference between WT and KO in the heterozygous deletion is not convincing because it appears to be accounted for by a few outliers, and there is no discussion of why this difference is not seen with homozygous TrkB deletion.

We agree with the reviewer about the lack of subtype specificity of GNF5837, as this compound inhibits TrkA TrkB and TrkC all at around 10 nM. For pharmacological inhibition, GNF5837 has been shown to be capable of crossing the blood-brain barrier (Albaugh et al., 2012). Previous studies have shown that deleting either TrkB or TrkC had little effect on neuronal death in the dentate gyrus since both TrkB and TrkC are active in DG, and one of these two Trk receptors could compensate for the loss of the other. Only double deletion of TrkB and TrkC could have a profound effect on cell survival in the dentate gyrus (Minichiello and Klein, 1996). Interestingly, we observed that GNF5837 injection modestly reduced the Fezf2-GFP+ progenitor cell number (Kv1.1 WT vs. Kv1.1 WT/GNF5837, Figure 7C), indicating a pan-Trk receptor suppression in DG. However, the genetic deletion of TrkB had no additional effect on the clone size (Figure 7 E and F). We hypothesize that the TrkC may compensate for the loss of TrkB in the neural progenitor cells and prevent further cell death. On the other hand, a gain-of-function of the Trk receptors could have a profound effect on cell function, such as enhanced cell proliferation. This gain-of-function can be specifically nullified by genetic deletion without affecting the basal level of Trk receptor function, which could be maintained by TrkC receptor.

The distribution of clone size is skewed as shown in a previous study (see Figure 1C, D from Singh et al., 2015). Although we only used a single TdTomato marker to label the clones, combined with a low-dose tamoxifen injection, we obtained comparable clone size distribution (Figure 7E and F).

If the larger clone in *Kcna1*^-/-^;*Ntrk2*^Flox/+^ was caused by the outliers, we would expect the distribution of the clone size to otherwise be comparable. However, our Kolmogorov-Smirnov cumulative plots of the clone size showed that the curve for *Kcna1*^-/-^;*Ntrk2*^Flox/+^ deviated from those for the other 3 genotypes (Figure 7F), which is consistent with the notion that the overall distribution of the clone size in *Kcna1*^-/-^;*Ntrk2*^Flox/+^ was right-shifted rather than affected by the outliers. This statistical analysis supports our hypothesis that Kv1.1KO mice tend to have larger clones.

Experimental concerns1. Figure 1E: It is not clear what the authors measured and how they did it. The y-axis tells us something about distance from SGZ in percent. The interpretation in direction of increased proliferation is not unlikely, especially in context of figure 7. However, S-phase shortening ihas been done at postnatal day 10, a time point when BDNF expression is rather low and under developmental control. The data are not linked and therefore not fully conclusive at this point of investigation.

For Figure 1E, we measured the relative position of the MADM-labeled neurons in the dentate gyrus. We first drew a vertical line (perpendicular to the SGZ) from the MADM-labeled neuronal soma so that it spans the distance from the SGZ to the border between the granule cell layer and molecular layer. For quantification, We designated the SGZ as the zero reference point (0%) and the granule layer/molecular layer boundary as the maximal position (100%), the farthest location the granule neurons can migrate into.

As stated in the first section in Results, "developmentally-born neurons generated during late embryogenesis and early postnatal stages typically survive for more than two months before the onset of cell death; generation and maturation of these neurons take about one month, with those generated early in life located farther from the SGZ (Cahill et al., 2017; Dayer et al., 2003; Kerloch et al., 2019; Toni and Schinder, 2015)." We found that the majority of green Kv1.1KO neurons were positioned farther away from SGZ. This result further supports the idea that the increase of Kv1.1KO neurons detected in 3 months old but not one-month-old MADM mice were born during the postnatal period. We have revised the text to clarify this point.

Although at P10, the BDNF level is low and under developmental control, the expression level can still be regulated by intrinsic as well as external factors, such as early postnatal environmental enrichment (Liu et al., 2012). Moreover, the Kv1.1 affects the membrane potentials that could directly regulate the BDNF releasing probability without affecting its mRNA expression, so the shortening of the S phase can still be observed at P10. Another reason we chose P10 as the age to investigate the S phase duration is we would like to avoid seizure attacks starting a week later, which could confound the study.

Nevertheless, to keep our manuscript concise and focused on the main point, we decided to remove Figure 8 and focus more on the BDNF/TrkB and electrophysiology.

2. Figure 2: In Figure 2, we expect to see that Kv1.1 is expressed in the dentate gyrus and is present in Fezf2-positve cells. The images are not easy to interpret. Overview images are missing. Where are these in situ dots? Which controls were used and how did the controls look like? I suppose that these are the progenitors close to the SVZ. Later in their study, Kv1.1 becomes functionally important in type2b cells or MADM-labelled cells within the DGin Fezf2-GFP positive cells. This image does not clarify whether Kv1.1 is also expressed in these cells, albeit it is very likely. Furthermore, immunolabeling of Kv1.1 minus mice (IHC) is missing. IHC, using this protocol, should be able to check immunoreactivity in the type2b cell type. This is important, as the authors provide evidence for changes in electrophysiology in type2b cells, but not the other cells that also express Kv1.1.

We have added new figures with overview and higher magnification images (Figure 2 A and B). We used the RNAscope technique for in situ detection of *Kcna1* mRNAs. The RNAscope technique is sensitive enough to detect a single mRNA molecule, and each dot represents one single mRNA molecule. Given the minuscule size of the mRNA dots, we have to show high-magnification images of each sample. The SGZ area is labeled with GFP granules (Fezf2-GFP), and numerous red *Kcna1* mRNA signals colocalize in the same area, indicating that the Kv1.1 is expressed in the Fezf2-GFP positive neural progenitor cells. We also performed the same experiments using Fezf2-GFP;*Kcna1*^-/-^ as the negative control (Figure 2B). The green GFP granules are still presented in the SGZ area; however, the red *Kcna1* mRNA signals are largely absent, thus providing evidence for the specificity of the *Kcna1* mRNA signal in Figure 2A.

We also performed IHC to detect the expression pattern of the Kv1.1 proteins. As shown in Figure 2C, Kv1.1 protein is expressed in SGZ and the granule layer of DG in control. Specifically, Kv1.1 immunoreactivity resides in doublecortin positive late-stage neural progenitor cells (white arrows), which largely overlap with type 2b cells. We found a very weak or undetectable Kv1.1 signal in Sox 2 positive cells. As expected, we also observed strong cytosolic Kv1.1 expression in DG neurons, which could be PV positive interneurons. This Kv1.1 protein signal is completely absent in Kv1.1KO mice. Interestingly, we noticed that the Sox2 signals were also detected in 2 out of 3 presumed inhibitory interneurons (yellow arrows). This phenomenon has been reported previously, indicating that Sox2 also can be detected occasionally in mature neurons (Kang and Hébert, 2012).

3. For Figure 2D-G, in order to support the conclusion that there are functional Kv1.1 channels in these progenitor cells, it is essential to present the raw un-subtracted current traces before and after Dtx-K application.

We have added raw data (Figure 2D to J) to show the un-subtracted current traces before and after DTX-K application. As shown in Figure 2 D to F, DTX-K sensitive currents can be seen in wild-type but not Kv1.1KO progenitor cells.

4. Figure 3C shows that there is a higher proportion of type 2b cells compared to other subtypes in the Kv1.1 knockout. This is interpreted as increased cell proliferation in the absence of Kv1.1. However, since Type 2A cells differentiate to Type 2b, a higher proportion of Type 2b cells could instead be accounted for by enhanced differentiation. This issue is not addressed.

If the differentiation from 2a to 2b is enhanced, we would expect to see relatively fewer 2a cells in Kv1.1KO mice. However, during our patch-clamp recording of the Fezf2-GFP positive cells, the chances of hitting type 2a and type 2b were comparable in both wild-type and Kv1.1KO mice (n for type 2a and 2b cells were 14 and 9 for wild-type and 14 and 10 for Kv1.1KO; p>0.05, Chi-square test). If the type 2a to 2b differentiation is enhanced in Kv1.1KO mice, it ought to reduce the probability of finding record type 2a cells for recording. Nevertheless, the possibility for enhanced differentiation cannot be completely ruled out.

5. Figure 3D shows that in Type 2b cells specifically, KV1.1 KO cells had a more depolarized RMP than WT. It is not clear if a sufficient number of cells was recorded for each group for this critical experiment, the figure legend reports a wide range of n=5 to 28 cells per group. The data in 3D are rather important, but are not well explained. Data should be presented as single data points. As the 2b cell type is in the focus of the analysis, a table with the cell type prediction data would help. The main effect is type 2b WT versus Kv1.1 KO, but a direct comparison of the data is missing. The authors should better explain why they used this statistical test (hypothesis?) and why they did the statistical comparison between the different groups.

We have added more cells and listed the n number for each group of cell type (n = 14, 14, 9, 10, 9 (WT) and 28, 14, 10, 11, 8) (Kv1.1KO) for type1 cells (Fezf2-GFP+/Sox2+), type 2a cells (Fezf2-GFP+/Sox2+/Tbr2+), type 2b cells (Fezf2-GFP+/Tbr2+), immature neurons (POMC-GFP+) and label-free mature neurons, respectively; (2-way ANOVA followed by Sidak's multiple comparisons test, p = 0.02, for type 2b cells), so sufficient sample sizes were used to perform the comparison for each cell type. We used scatter plots instead of single data points so the data distribution and skewness can be appreciated. The single data points of Figure 4G can also be seen in Figure 5 B and C.

The 3D plots (Figure 5 B and C) show the non-linearity among the 3 biophysical parameters for cell type prediction. Nevertheless, we still can observe a continuous transition from Type 1 cells through type 2a cells towards type 2b cells. The labeled cells are dispersed among the cells with unknown cell identity, and therefore, so we decided to use the Multinomial logistic regression. This simplified machine-learning method can handle datasets with a relatively smaller sample size. The cell type was determined based on the highest probability, and despite a smaller training dataset, we still obtained a reasonable accuracy rate (~80%), similar to the more complex AI-based machine learning method that handles a much larger dataset.

We used 2 separate training datasets for classification. This would give us more stringent classification criteria, as Kv1.1KO type 2b cells were more depolarized (Figure 4G). Those cells in the Kv1.1KO group must reach more depolarized membrane potential to be qualified as type 2b cells. Despite that, we still clearly observed that more Kv1.1KO cells had more depolarized resting membrane potentials, and the majority of these cells also had smaller capacitance and higher input resistance, which allowed them to be classified as type 2b cells in the Kv1.1KO group that entails more stringent criteria.

6. It has not been shown whether the systemically applied Trk inhibitor affects Trk phosphorylation in the hippocampus, meaning across the blood brain barrier, in these experiments.

We found the intraperitoneal injection of GNF5837 completely suppressed the auto-phosphorylation of TrkB (Figure 6—figure supplement 2), suggesting our regimen was sufficient for GNF5837 to cross the blood-brain barrier and inhibit the TrkB signaling. Moreover, this result also demonstrated that TrkB was activated locally in the SGZ.

7. Clonal analysis in Figure 6A-C affects Fezf2-GFP, but the criteria for clonal analysis is not well described. By going through the methods I realized that this is the 3D analysis of clones using Td Tomato. Image data of this analysis should be presented and the way of analysis must be clear. As it is not easy to obtain unbiased analysis of clone sizes with such an approach, it is important to show how it has been done and how the images look like.

We defined a single clone by including all the TdTomato-positive cells contained within a 100-μm radius of the clone center. We added Figure 7—figure supplement 1 to illustrate how the clones were identified and quantified. During this revision, we tried to generated Gli1-creER;*Kcna1*;MADM6 mice so the clones can be better identified by 3 colors instead of a single color and the resolution can be increased as well. Unfortunately, we have tried 3 different tamoxifen regimens and analyzed more than 20 mice, only one single clone was found. We suspect the rare inter-chromosomal recombination of MADM plus the low-efficiency of tamoxifen-inducible Gli1-cre rendered this inducible MADM not feasible for clonal analysis.

8. The most critical point is that knocking out TrkB from adult born neurons is sufficient to interfere with neurogenesis (e.g. Bergami et al. PNAS 2008) and this effect was not be connected to Kv1.1. So the links between BDNF, TrkB, and more division at early stages are not causally connected. The overall idea is that Type2b cells that lack Kv1.1 are briefly depolarized, that this causes local BDNF release and a stimulation of cell division. However, it has not been shown that these cells, or close neighbors express more BDNF, that this stimulates TrkB and that this specific cell type causes the increase in adult born neuron numbers. Furthermore, it may be that the increase in the RMP does not lead to a higher activity of the neuron and activity-dependent gene expression of BDNF, because an increase in this parameter may induce a depolarization block; unlikely, but not ruled out and a relevant problem in excitability research.

As described by the reviewer, BDNF has a wide range of expression patterns in the dentate gyrus, including both mature neurons and the neural progenitor cells in SGZ (Minichiello and Klein, 1996). Moreover, previous studies have also demonstrated an increase of BDNF mRNA expression in SGZ in *megencephaly* mice (Lavebratt et al., 2006). For these reasons, attempting to identify the origin of BDNF that activates the TrkB signaling in SGZ has become a daunting task. BDNF acts as a short-range neuromodulator, and just like other neurotransmitters and neuromodulators, the release of BDNF is calcium-dependent, which mostly requires membrane depolarization that activates calcium channels (Mandl et al., 2002). Our electrophysiological recording results show that only Type 2b cells were more depolarized in Kv1.1KO mice. Even though we cannot exclude the possibility that neurons could also serve as another source of BDNF, our clonal analysis and MADM results do not support this idea. For a more definitive assessment, we will need more sophisticated and sensitive methods to detect the BDNF release from different cell types, but these tools are still under development (Inutsuka et al., 2021).

Albaugh, P., Fan, Y., Mi, Y., Sun, F., Adrian, F., Li, N., Jia, Y., Sarkisova, Y., Kreusch, A., Hood, T., et al. (2012). Discovery of GNF-5837, a Selective TRK Inhibitor with Efficacy in Rodent Cancer Tumor Models. ACS Medicinal Chemistry Letters 3, 140-145.

Almgren, M., Nyengaard, J.R., Persson, B., and Lavebratt, C. (2008). Carbamazepine protects against neuronal hyperplasia and abnormal gene expression in the megencephaly mouse. Neurobiol Dis 32, 364-376.

Inutsuka, A., Ino, D., and Onaka, T. (2021). Detection of neuropeptides in vivo and open questions for current and upcoming fluorescent sensors for neuropeptides. Peptides 136, 170456.

Kang, W., and Hébert, J.M. (2012). A Sox2 BAC Transgenic Approach for Targeting Adult Neural Stem Cells. PLoS ONE 7, e49038.

Lavebratt, C., Trifunovski, A., Persson, A.S., Wang, F.H., Klason, T., Ohman, I., Josephsson, A., Olson, L., Spenger, C., and Schalling, M. (2006). Carbamazepine protects against megencephaly and abnormal expression of BDNF and Nogo signaling components in the mceph/mceph mouse. Neurobiol Dis 24, 374-383.

Li, G., Fang, L., Fernández, G., and Pleasure, S.J. (2013). The ventral hippocampus is the embryonic origin for adult neural stem cells in the dentate gyrus. Neuron 78, 658-672.

Liu, N., He, S., and Yu, X. (2012). Early Natural Stimulation through Environmental Enrichment Accelerates Neuronal Development in the Mouse Dentate Gyrus. PLoS ONE 7, e30803.

Mandl, S., Sader, R., Thorwarth, G., Krause, D., Zeilhofer, H.F., Horch, H.H., and Rauschenbach, B. (2002). Investigation on plasma immersion ion implantation treated medical implants. Biomol Eng 19, 129-132.

Minichiello, L., and Klein, R. (1996). TrkB and TrkC neurotrophin receptors cooperate in promoting survival of hippocampal and cerebellar granule neurons. Genes and Development 10, 2849-2858.

Rho, J.M., Szot, P., Tempel, B.L., and Schwartzkroin, P.A. (1999). Developmental Seizure Susceptibility of Kv1.1 Potassium Channel Knockout Mice. Developmental Neuroscience 21, 320-327.